



# Learning Landscape Features from Streamflow with Autoencoders

Alberto Bassi[1,2], Marvin Höge[2], Antonietta Mira[3,4], Fabrizio Fenicia[2], and Carlo Albert[2]

[1]Department of Computer Science, ETH Zurich, Switzerland
[2]Swiss Federal Institute for Aquatic Science and Technology (Eawag), Dübendorf, Switzerland
[3]Euler Institute, Università della Svizzera italiana, Lugano, Switzerland
[4]Insubria University, Como, Italy

**Correspondence:** Alberto Bassi (abassi@ethz.ch)

**Abstract.**

Understanding the number and types of signatures that best describe streamflow time series is a crucial objective in hydrological science, serving applications such as catchment classification, hydrological model development and calibration. With the main objective of learning a minimal number of streamflow features, we employ an explicit noise conditional autoencoder

(ENCA), which, together with meteorological forcings, allows for an accurate reconstruction of the whole streamflow time series. The ENCA architecture feeds the meteorological forcing to the decoder in order to incentivize the encoder to only learn features that are related to landscape properties. By isolating the effect of meteorology, these features can thus be interpreted as landscape fingerprints. The optimal number of features is found by means of an intrinsic dimension estimator. We train our model on the hydro-meteorologic time series data of 568 catchments of the continental United States from the CAMELS

dataset. We compare the reconstruction accuracy with state-of-the-art models that take as input a subset of static catchment attributes (both climate and landscape attributes) along with the meteorological forcing variables. Our results suggest that available static catchment attributes compiled by experts account for almost all the relevant information about the rainfall-runoff relationship. Yet, these catchment attributes can be summarized by only two relevant learnt features (or signatures), while a third one is needed for about a dozen difficult catchments in the central US, mainly characterized by high aridity index and

intermittent flow. The principal components of the learnt features strongly correlate with the baseflow index and aridity indicators, which is consistent with the idea that these indicators capture the variability of catchment hydrological response and relate to needed model complexity. The correlation analysis further indicates that soil-related and vegetation attributes are of high importance. Finally, in the attempt to interpret the learnt catchment features, we relate them to typical hydrological model components, with specific reference to the parameters of the GR4J model and their function on the hydrograph.

## 1 Introduction

*Hydrological signatures* encompass descriptive statistics derived from meteorological and streamflow time series. They serve various purposes in hydrology, such as hydrological model calibration or evaluation (Fenicia et al., 2018; Kiraz et al., 2023), process identification (McMillan, 2020a), and ecological characterization (Olden and Poff, 2003). Alongside with catchment





attributes (distinguished here in landscape and climate attributes), they are also used for catchment classification and regional-
ization studies (Wagener et al., 2007).

*Streamflow signatures*, i.e. hydrological signatures solely based on streamflow, hold significant importance as they relate to
the variable one aims to predict and understand (Gnann et al., 2021). Hydrologists have developed diverse signatures reflecting
different aspects of streamflow dynamics. Examples include those linked to the flow duration curve (e.g., slope of various
segments), the baseflow index, or the flashiness index. Numerous other such signatures exist. For instance, Olden and Poff
(2003) compiled a list of 171 indices from prior work, reflecting aspects associated to magnitude, frequency, duration, timing,
and rate of change of flow events. As streamflow depends on meteorological forcing and landscape attributes, streamflow
signatures generally contain information from both sources. In particular for predictions in ungauged basins, it is vital to be
able to disentangle them.

One way of doing so is through *hydrological models*, which condense catchment attributes into *model parameters* (Wagener
et al., 2003). Previous research indicates that observed hydrographs can be represented by a handful of model parameters
(Jakeman and Hornberger, 1993). For instance, the *GR4J model* (Perrin et al., 2003), resulting from a continuous refinement
process aimed at balancing model complexity and performance, has only four parameters. However, these analyses are based
on predefined model assumptions.

Model parameters can, in principle, directly be estimated from streamflow signatures. The Approximate Bayesian Computa-
tion (ABC) technique (Albert et al., 2014) has recently been used to infer model parameters from streamflow signatures - which
in this context are called *summary statistics* - bypassing the need to directly compare the complete time series (Fenicia et al.,
2018). If these summary statistics contained all the information necessary to estimate model parameters they would be termed
as *sufficient*. Sufficiency is therefore not an inherent property of the summary statistics but depends on the specific hydrological
model and on the parameters that need to be inferred. For ABC to converge efficiently, we also want the summary statistics
to be *minimal*, i.e., while they should ideally encode all the parameter related information available in the streamflow, they
should encode no other information, neither from the forcing nor from the noise that is used for the simulations (Albert et al.,
2022). Such minimally sufficient summary statistics could thus be considered the relevant fingerprints of landscape features
on the streamflow. Of course, this holds true only if the model is capable of encoding all the information in such features that
is relevant for the input-output relationship. However, recent studies show that purely data-driven models might outperform
process-based models in prediction accuracy (Kratzert et al., 2019; Mohammadi, 2021), becuase they suggest information in
catchment attributes previously not utilized for streamflow prediction.

Our goal is to employ Machine Learning (ML) techniques to identify a minimal set of streamflow features enabling accurate
streamflow predictions when combined with meteorological forcing. Thus, our aim is to eliminate all forcing-related informa-
tion from the streamflow, distilling features solely from the catchments themselves. We approach this objective from a purely
data-driven perspective, aiming to reduce assumptions relative to traditional process-based modeling.

To identify minimal sets of streamflow features, we employ a novel ML architecture recently proposed for extracting mini-
mally sufficient summary statistics from noisy outputs of stochastic models. The architecture is an Explicit Noise Conditional
Autoencoder (ENCA) (Albert et al., 2022), where the bare noise utilized by the stochastic model simulator is fed into the de-





coder together with the learnt summary statistics. This way, the encoder is encouraged to encode only those features containing

information on the model parameters while disregarding the noise. Albert et al. (2022) applied ENCA to infer parameters of simple one-dimensional stochastic maps, showing that the learnt features allow for an excellent approximation of the true posterior. In our case, instead of noise, we input meteorological forcing into the decoder and encourage the encoder to exclusively encode landscape-related information within the streamflow. In order to reduce the computational costs and learn a minimal set of catchment features, the dimension of the latent space is chosen according to the estimation of its Intrinsic Dimension

(ID) (Facco et al., 2017; Allegra et al., 2020; Denti et al., 2022). In particular, we employ the ID estimator GRIDE (Denti et al., 2022), which is robust to noise. Learnt features will then be compared with known catchment attributes (both from the landscape and the climate) and hydrological signatures to provide a hydrological interpretation and guide knowledge domain experts towards the pertinent information necessary for streamflow prediction.

We apply our approach to the US-CAMELS dataset (Newman et al., 2015), covering several hundred catchments over the

continental US. In order to benchmark our results, we used previous modelling work on the same dataset. LSTMs (Long Short Term Memory networks) have emerged as state-of-the-art models for data-driven predictions in ungauged basins. In the study of Kratzert et al. (2019), LSTMs validated on unseen catchments, enriched with static landscape and climate attributes from Addor et al. (2017), outperformed conceptual models. First investigations towards mechanistic interpretation of the LSTM states, e.g. linking hidden states to the dynamics of soil moisture, demonstrated the potential of eliciting physics from data-

driven models (Lees et al., 2022). Here, by linking learnt features to known catchment attributes, we explore a further aspect in this broader field of explainable AI or interpretable ML (Molnar, 2024; Molnar et al., 2020).

Our specific objectives are: (i) find the minimal number of dominant streamflow-features stemming from the landscape; (ii) relate them to known landscape and climate attributes as well as established hydrological signatures. This will not only allow us to determine how many features are required for streamflow prediction, but also to answer the question whether there is

missing information in known catchment attributes.

A similar attempt of learning signatures has recently been made by Botterill and McMillan (2023). In pursuit of an interpretable latent space on a continental scale, they employed a convolutional encoder to compress high-dimensional information derived from meteorological forcing and streamflow data. This approach was aimed at learning hydrological signatures (McMillan, 2020b) within the US-CAMELS dataset. Their approach differs from ours in three aspects: (i) they used a a tradi-

tional conceptual model as a decoder whereas we use an LSTM architecture which has been shown to be superior to conceptual models when provided with catchment properties; (ii) they fed both streamflow and meteorological forcing into the encoder whereas we feed in only streamflow data in an attempt to separate landscape- from forcing-information; (iii) they did not attempt to find a minimal number of signatures sufficient for streamflow prediction, whereas this is a primary objective of our work.

It is important to note that our main objective is not to beat state-of-the-art models regarding their predictive performance in ungauged basins. Our goal is rather to investigate the information content in streamflow. However, we believe our research will provide valuable insights into the most critical features for streamflow prediction. Ultimately, this knowledge may be utilized for prediction in ungauged catchments in the future.





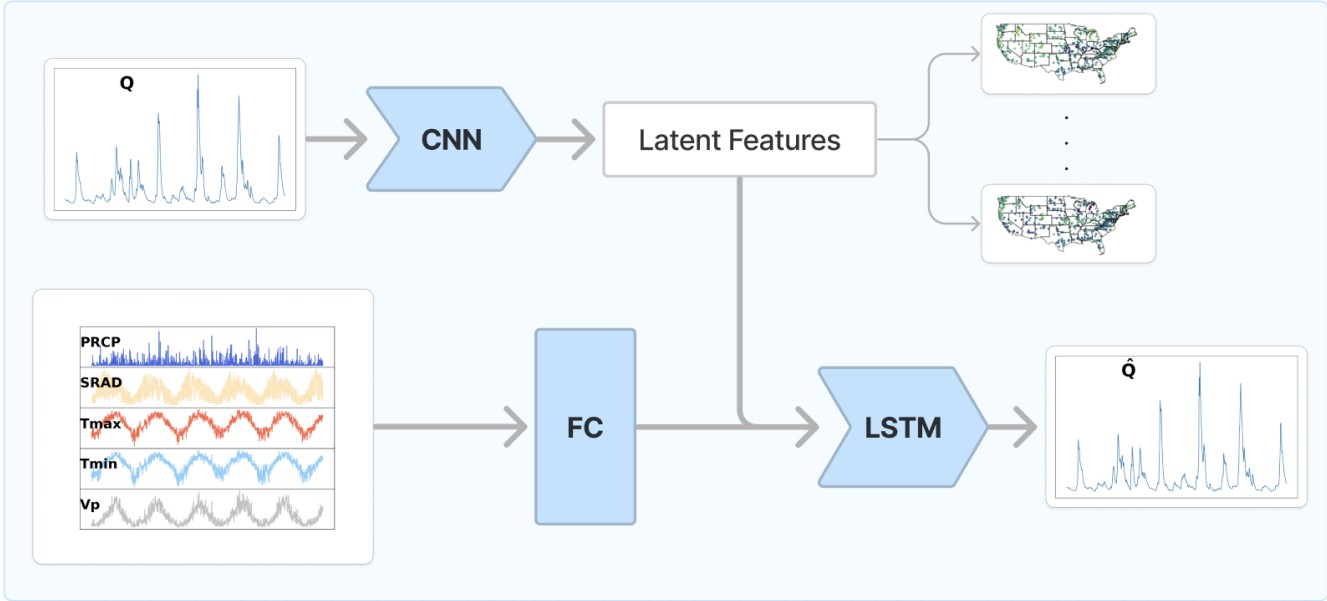

**Figure 1.** Explicit Noise Conditional Autoencoder used in this study. For the hyper-parameters and the implementation details, the reader is referred to Appendix C. The neural network architectures employed are Convolutional Neural Networks (CNN), a Fully Connected (FC) layer and LSTM. The observed and simulated streamflows are denoted with $Q$ and $\hat{Q}$, respectively. The meteorological forcing variables are denoted with PRCP (precipitation), SRAD (solar radiation), Tmax (maximum temperature), Tmin (minimum temperature) and Vp (vapour pressure).

## 2 Models and Methods

### 2.1 Data

We employ the Catchments Attributes and Meteorology for Large-sample Studies (CAMELS) (Newman et al., 2015), which consists of 671 catchments in the contiguous United States (CONUS), ranging in size from $4$ to $25 \cdot 10^3$ km$^2$. For this study, we select those 568 catchments out of 671 whose data span continuously on a daily basis the time period from 1 October 1980 to 30 September 2010, corresponding to 30 hydrological years. The first 15 years are used for calibration and the last 15 for validation. Along with the streamflow time series and the meteorological forcing variables, US-CAMELS also provides information about catchments static attributes (Addor et al., 2017), encompassing both landscape (vegetation, soil, topography and geology) and climate. Streamflow data is retrieved from the U.S. Geological Survey gauges, while the meteorological forcing comes from the North America Land Data Assimilation Systems (NLDAS) and includes maximum and minimum daily temperature, solar radiation, vapour pressure and precipitation.



## 2.2 Explicit Noise Conditional Autoencoder

We use the Explicit Noise Conditional Autoencoders (ENCA) (Albert et al., 2022), where the streamflow is fed to a convolutional encoder. ENCA has been designed to distill sufficient summary statistics which contain minimal noise information from the output of stochastic models. Here, the noise is substituted by all the variables we are not interested in, namely the meteorological forcing. The convolutional encoder is thus followed by a LSTM decoder that takes as input 15 hydrological years of meteorological forcing, i.e. 5478 values (Figure 1 - for the detailed architectures the reader is referred to Appendix C). The memory cells of the LSTM are limited by the dimension of the input layer. In order to enlarge the memory available and capture more complex patterns from the meteorological forcing, the meteorological time series are first fed to a single linear layer with 1350 output units. The output of this linear layer is then concatenated with the output of the encoder and fed to the LSTM decoder. This way the decoder sees tensors of size $(B, 5478, 1350 + S)$, where $B$ is the batch size (batches are selected across different catchments) and $S$ is the latent space dimension. We opted for such an architecture in order to extract as much static information related to the streamflow as possible.

Obviously, setting the latent dimension equal to the length of the entire time-series would allow for a perfect streamflow reconstruction. However, we expect to be able to compress almost all information not already contained in the forcing into a much smaller number of features. Because they should be largely devoid of forcing information, we call them the *relevant landscape features* and explain in the next section how we fix their number. We refer to the ENCA model with latent space dimension equal to $N$ as ENCA-$N$. Comparing relevant landscape features with known static catchment attributes in terms of their capacity for streamflow reconstruction will allow us to find out whether static catchment attributes lack information that is crucial for streamflow prediction. For this comparison, we use an LSTM model augmented with catchment attributes (Addor et al., 2017) in the input. We refer to this model as Catchment Attributes Augmented Model (CAAM). This model differs from Figure 1 solely by the fact that the latent features are substituted by known catchment attributes. Following Kratzert et al. (2019), CAAM is fed with 27 catchment attributes (reported in Table 1 and denoted with a *), which are representative of climate, topology, geology, soil and vegetation. As a control case, we also report the results obtained with ENCA-0, which is given neither catchment attributes nor learnt features as inputs.

In order to mitigate numerical instability, it is crucial to standardize the catchment attributes or latent features before feeding the LSTM. In CAAM, we standardize the catchment attributes once and for all with the mean and standard deviation computed over all the considered catchments. This is not possible for ENCA, since the mean and standard deviation of the latent features are not known a priori. Therefore, we standardize the latent features by mean of a batch normalization layer. This way, we ensure that the magnitude of the LSTM input is comparable between CAAM and ENCA.

## 2.3 The Intrinsic Dimension

To identify the dimension of the latent space we proceed with the following methodology. First, we train an ENCA-$N$ with a relatively large number of latent features $N$. Since we fed 27 catchment attributes to the reference model (CAAM), we used a 27-dimensional latent space in order to have a fair comparison in terms of model capacity. We refer to this model as ENCA-27.

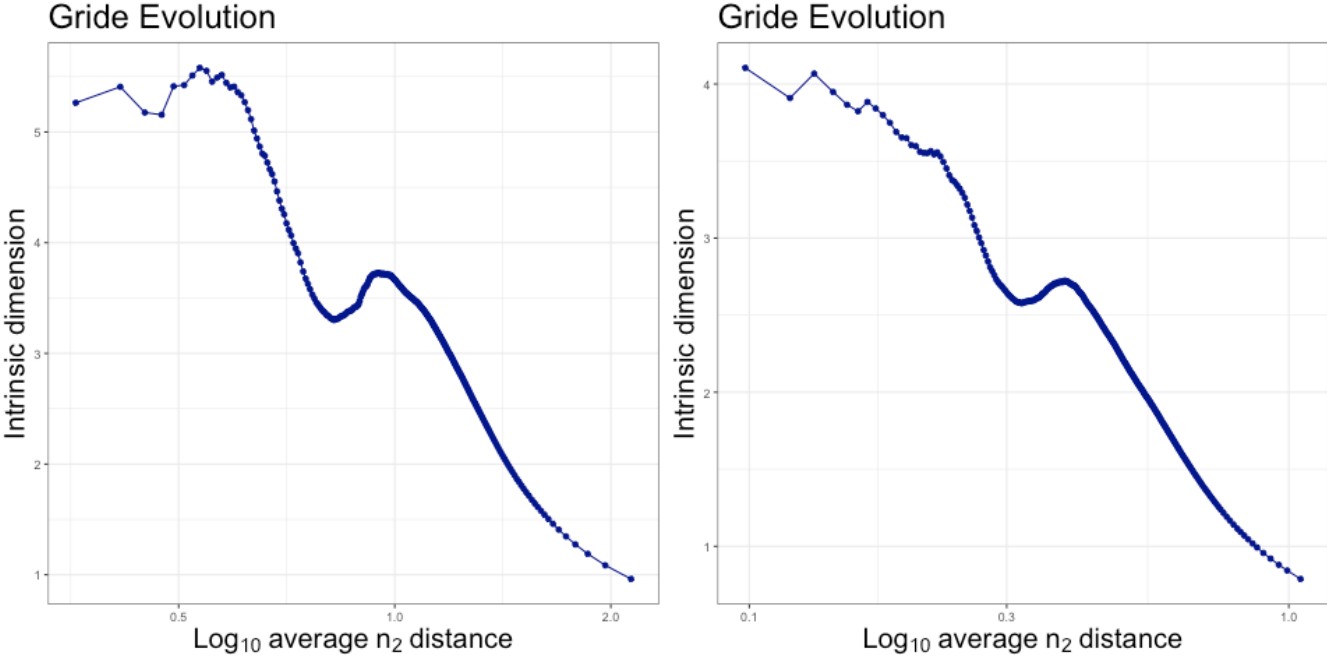

**Figure 2.** GRIDE evolution plot for the ENCA-27 (left) and the ENCA-5 (right) latent spaces for one random restart. The other restarts show a similar pattern.

The dimension of the latent space does not matter so much, as long as it is larger than the expected number of relevant landscape features. Then, we estimate the ID of the latent space (see below), and train another ENCA with the number of latent features equal to the estimated ID and, in turn, estimate its ID to check if the dimension of the latent space can be further reduced.

In order to estimate the ID, we apply the GRIDE estimator (Denti et al., 2022). Given sample points, $\mathbf{x}_i \in \mathbb{R}^D$, for $i = 1, \ldots, M$, and a distance measure, $r : \mathbb{R}^D \times \mathbb{R}^D \to \mathbb{R}^+$, GRIDE assumes that points in a given neighbourhood are counted with a Poisson point process with intensity $\rho$, which is constant at least up to the scale of the diameter of the considered neighbourhood. We define $r_{i,l}$ as the distance between the point $\mathbf{x}_i$ and its $l$-th nearest neighbour. Let us define $\mu_{i,n_2,n_1} = \frac{r_{i,n_2}}{r_{i,n_1}}$, where $n_1$ and $n_2$ (with $0 \leq n_1 \leq n_2 \leq M$) are nearest neighbours of generic order. The distribution of $\mu_{i,n_2,n_1}$ can be computed in closed form and depends only on the ID of the data while, crucially, does not depend on $\rho$, as long as $\rho$ is constant in the considered neighbourhood of $i$ whose diameter is set by the distance between $i$ and its $n_2$-th nearest neighbour (Denti et al., 2022).

In order to correctly identify the ID of a dataset, a scale-independent analysis is essential. We therefore make use of *GRIDE paths*, the evolution of the ID as a function of $n_2$, which can be interpreted as the scale at which we look at the data. We set $n_1 = n_2/2$, as is usually done in the literature. As a function of $n_2$, the ID is first expected to increase, due to the noise present at small distance scales, and then to reach a plateau corresponding to the correct ID. Figure 2 shows the GRIDE path for increasing values of $n_1$ from 1 to 270. The left panel shows that the ID estimate of the latent space of ENCA-27 decreases





after showing a plateau around five, then reaches a minimum around three, then increases again and finally collapses to low
values at larger distance scale. The plateau at five motivates us to train an ENCA-5 and study its ID. The right panel of Figure 2
shows that the local minimum of the GRIDE path of the latent space of ENCA-5 is consistent with an ID of three. We can
deduce that, for most of the catchments, the ID is three, while for some it can be higher. However, the fact that the GRIDE path
of the latent space of ENCA-27 shows two plateaus around five and three can be an indicator of the existence of two or more
manifolds with different IDs. We will see below that, indeed, three features seem to capture most of the relevant information,
which is in line with the GRIDE path for ENCA-5 (right plot in Figure 2).

Note that, at increasingly large scales, the GRIDE estimator is not reliable anymore since the assumption of locally homogeneous intensity of the Poisson process - on which it relies - may fail to hold. With real data it is usually difficult to ascertain the
scale at which the local homogeneity assumption is valid. We use the ID as a guide to reduce the dimension of the latent space
of the autoencoder. However, in the end we train ENCA for several dimensions of the latent space and evaluate the information
content of the learnt features in terms of their ability to reconstruct streamflows.

## 2.4 Training and Validation

We use the first 15 years of data for calibration and the last 15 for validation. Calibration is performed by maximizing the
Nash-Sutcliffe Efficiency (NSE) (Nash and Sutcliffe, 1970), defined as:

$$\text{NSE} = 1 - \frac{\sum_{t=1}^{T} \left(q_{\text{sim},t} - q_{\text{obs},t}\right)^2}{\sum_{t=1}^{T} \left(q_{\text{sim},t} - \mu_{obs}\right)^2} \,, \tag{1}$$

where $q_{obs,t}$ and $q_{sim,t}$ are, respectively, the observed and predicted streamflow expressed in $mm/day$ at day $t$, and $\mu_{obs}$ is the
average of the observed streamflow. We notice that maximizing the NSE is equivalent to minimizing the Mean Square Error
(MSE) between data and prediction. Each model is trained with the Adam optimizer (Kingma and Ba, 2015), with learning
rate equal to $10^{-5}$ for a maximum of $20 \cdot 10^3$ epochs and early stopping with patience of $2 \cdot 10^3$ epochs. The models with best
validation NSE is then chosen. The batch size is set to 64 and the first 270 days of the predicted streamflow are excluded when
computing the loss.

Since the NSE overweight flow peaks due to the square values, it is not well suited to asses the performance on low flow
regimes. Therefore, following Kratzert et al. (2019) we also report the percentage bias, defined as

$$\text{BIAS} = \frac{\mu_{\text{sim}} - \mu_{\text{obs}}}{\mu_{\text{obs}}} \,. \tag{2}$$

In addition, to assess the performance in streamflow variability, we report the standard deviation ratio of the streamflow
logarithm, defined as

$$\text{LOG} - \text{STDEV} = \frac{\sigma_{\log(\text{sim})}}{\sigma_{\log(\text{obs})}} \,, \tag{3}$$

where $\sigma_{\log(\text{sim})}$ and $\sigma_{\log(\text{obs})}$ are the standard deviations of the logarithm of the simulated and observed streamflows, respectively.





Each algorithm is affected by noise, due to the random initialization of the neural network parameters. To minimize this
effect we run each model with four random restarts, each one providing the streamflow prediction in the whole validation
period. We compute the evaluation metrics on the predicted streamflow after averaging the streamflow over these four random
restarts.

## 2.5 Latent Space Interpretation

The relevant features are first projected using a Principal Component Analysis (PCA), since in general the autoencoder latent
representation is in arbitrary coordinates. By doing this, we ensure a fair comparison between different random restarts, since
we change the basis of each latent space by ordering the new coordinates according to the explained variance. Finally, in order
to interpret the relevant landscape features, we report the Spearman correlation (Zar, 2005) matrix between the learnt features
and static catchment attributes and hydrological signatures, which are reported in Table 1.





| Meteorological Forcing Variables | |
|---|---|
| PRCP | Average daily precipitation ($mm/day$) |
| SRAD | Surface incident solar radiation ($W/m^2$) |
| Tmax | Maximum daily atmosphere temperature (°C) |
| Tmin | Minimum daily atmosphere temperature (°C) |
| Vp | Nearly surface daily vapour pressure average (Pa) |
| **Climate Attributes** | |
| Prec Mean* | Mean daily precipitation. |
| PET Mean* | Mean daily potential evapotranspiration. |
| Prec Seasonality* | Seasonality of precipitation estimated by using sinusoidal waves. |
| Fraction Snow* | Fraction of precipitation falling on days with T < 0 °C. |
| Aridity Index* | Ratio between the mean PET and mean precipitation. |
| High Prec Frequency* | Frequency of days with ≤ 5x mean daily precipitation. |
| High Prec Duration* | Mean duration of high precipitation events. |
| Low Prec Frequency* | Frequency of days with ≤ 1 mm/day of precipitation. |
| Low Prec Duration* | Mean duration of dry periods. |
| streamflow Ratio | Ratio between the mean daily streamflow and mean daily precipitation. |
| Stream ELAS | Streamflow precipitation elasticity. |
| **Hydrological Signatures** | |
| Q Mean | Mean daily streamflow (mm/day). |
| Slope FDC | Slope of the flow duration curve. |
| Baseflow Index | Ratio between the average daily baseflow and streamflow. |
| Q5 | 5 % flow quantile (mm/day). |
| Q95 | 95 % flow quantile (mm/day). |
| High Q Frequency | Frequency of high-flow days (> 9 times the median daily flow). |
| High Q Duration | Mean duration of high flow events (number of consecutive days > 9 times the median daily flow). |
| Low Q Frequency | Frequency of low flow days (< 0.2 x the mean daily flow). |
| Low Q Duration | Mean duration of low-flow events (number of consecutive days < 0.2 times the mean daily flow). |
| Zero Q Frequency | Frequency of days with streamflow = 0 $mm/day$. |
| HFD Mean | Mean half-low-date (date on which the cumulative streamflow since October the 1st reached half of the annual streamflow). |
| **Landscape Attributes** | |
| *Topological Attributes* | |
| Elevation Mean* | Mean elevation of the catchment. |
| Slope Mean* | Mean slope of the catchment. |
| Area Catchment* | Area of the catchment. |
| *Geological Attributes* | |
| Carbonate Rocks Fraction* | Carbonate sedimentary rocks fraction area in the catchment. |
| Geological Porosity | Subsurface porosity. |
| Geological Permeability* | Surface permeability (in log10 scale). |
| *Soil Attributes* | |
| Soil Depth (Pelletier)* | Depth to bedrock (maximum 50 m). |
| Soil Depth (STATSGO)* | Soil depth (maximum 1.5 m). |
| Soil Porosity* | Volumetric porosity. |
| Soil Conductivity* | Saturated hydraulic conductivity. |
| Max Water Content* | Maximum water content of the soil. |
| Sand Fraction* | Fraction of sand in the soil. |
| Silt Fraction* | Fraction of silt in the soil. |
| Clay Fraction* | Fraction of clay in the soil. |
| Water Fraction | Fraction of the top 1.5 m marked as water. |
| Organic Fraction | Fraction of the soil depth (STATSGO) marked as organic material. |
| Other Fraction | Fraction of the soil depth (STATSGO) marked as other. |
| *Vegetation Attributes* | |
| Fraction Forest* | Fraction of the catchment covered by forest. |
| LAI Max* | Maximum monthly mean of leaf area index. |
| LAI Diff* | Difference between the max. and the min. monthly mean of the leaf area index. |
| GVF Max* | Maximum monthly mean of the green vegetation fraction. |
| GVF Diff* | Difference between the max. and min. monthly mean of the green vegetation fraction. |
| Root Depth 50 | 50 % percentile extracted from a root distribution based on IGBP land cover |
| Root Depth 99 | 99 % percentile extracted from a root distribution based on IGBP land cover |

**Table 1.** Meteorological forcing variables, climate and landscape (topological, geological, soil and vegetation) attributes and hydrological signatures compared in this study. The attributes fed to CAAM are denoted with a *.





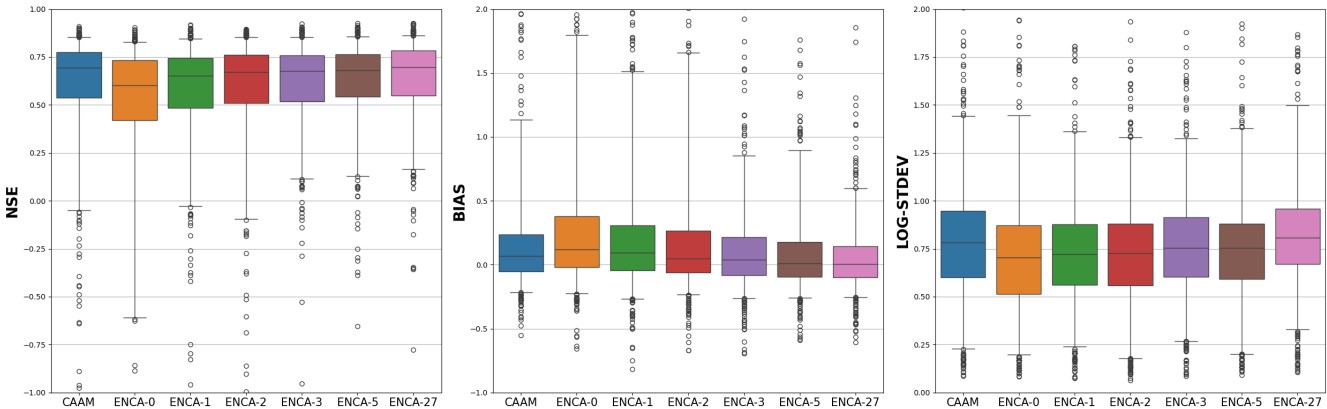

**Figure 3.** Validation NSE, BIAS and LOG-STDEV values for the considered models. The NSE and BIAS distributions of the attributes augmented model (CAAM) lie between ENCA with 2 and 3 latent features. ENCA tends to underestimate the flow variability, which approaches one by increasing the number of latent features.

## 3 Results and Discussion

### 3.1 The Number of Relevant Landscape Features

Figure 3 depicts the boxplot of the validation NSE, BIAS and LOG-STDEV values for the considered models. The associated statistics are reported in Appendix A. In all the metrics considered, the control model ENCA-0 achieves the worst results, which is consistent with what Kratzert et al. (2019) found. In particular, the big performance difference between ENCA-0 and CAAM is an indicator of the utility of the information contained in the 27 selected catchment attributes in terms of streamflow prediction.

In general, Figure 3 and the related statistics (Table A1) show also that increasing the number of latent features improves the prediction accuracy of the considered metrics. All the ENCA-$N$ models (for $N > 0$) perform better (with respect all the metrics considered) than the control case ENCA-0.

In terms of NSE, we observe a performance improvement from ENCA-1 to ENCA-2 in the bulk of the distribution, and a further improvement from ENCA-2 to ENCA-3 which produces a lower number of NSE outliers. The NSE improvements between ENCA-3 and ENCA-27 are minor both in the outliers and in the bulk distribution. In terms of the BIAS, we observe a similar pattern. By increasing the number of latent features, the bulk distribution improves significantly. However, we observe the biggest gap between ENCA-2 and ENCA-3 and this gap is mostly related to high BIAS outliers. Overall, CAAM performance is most similar to ENCA-2 in terms of BIAS and NSE. We therefore argue that catchment attributes collected by experts account for two relevant landscape features that appear to be sufficient for most catchments, while at least a third one is needed to resolve specific catchments.





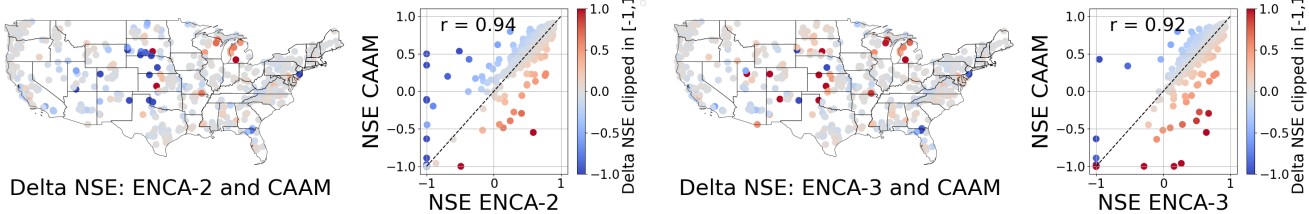

**Figure 4.** Validation NSE of ENCA-2 (left panel) and ENCA-3 (right panel) versus CAAM, color-coded with the NSE difference per catchment clipped in [-1,1]. Red means ENCA performs better, blue means CAAM performs better.

Finally, regarding LOG-STDEV, it is evident that both CAAM and ENCA models tend to underestimate the streamflow variability. The results of ENCA-27 are most similar to CAAM while all other ENCA variants result in lower LOG-STDEV values. We hypothesize that ENCA models perform worse than CAAM in LOG-STDEV because of the different standardization pro-
cedure employed. However, the general underestimation of streamflow variability was also described in other LSTM-based hydrological modelling investigations (e.g. Kratzert et al., 2019). It can partly be attributed to using NSE as objective function for training which puts more weight on matching high flow. Also, using daily averaged data means covering faster dynamics and variability in the system. In addition, LSTM models that are trained on these data might not capture all the dynamics of even this daily data using the static attributes in CAMELS - that themselves are averages over entire catchments. The ENCA
results show lower but increasing LOG-STDEV values indicating that using more latent features might help to better match the hydrograph variability. At the same time, ENCA-27, that would be "free" to learn whatever feature it deems necessary to be encoded to match the hydrograph variability, does now exceed the CAAM performance in this respect and this might confirm the existence of an upper bound to the LSTMs performance.

To study which catchments are most affected when using the latent features of the ENCA models in place of the known
catchment attributes in CAAM, we report (Figure 4) the NSE difference between CAAM and ENCA-2 (left panels) or ENCA-3 (right panels), respectively. While, for most catchments, switching from ENCA-2 to ENCA-3 does not result in a high performance gain, we see a clear improvement on about a dozen or so catchments, mostly located in the central CONUS. This corroborates the hypothesis that the collected catchment attributes account for two relevant landscape features and the improvement due to the third one is related to only few catchments that are particularly difficult to predict. It is interesting
to count the number of catchments for which CAAM fails (negative NSE values, i.e. predictions that are worse than average streamflow) but ENCA succeeds (positive NSE values). This number increases from 13 (ENCA-2) to 21 (ENCA-3). On the other hand, there are only 9 (3) catchments, for which CAAM succeeds but ENCA-2 (-3) fails.

In order to evaluate the impact of additional features, we compare the performances of ENCA models differing in their number of latent variables (Figure 5). The results corroborate our earlier findings that two features are sufficient to cover most
of the catchments, and additional features provide information about relatively few, difficult to predict, mostly catchments in the central CONUS dominated by arid climate conditions. While the number of such catchments informed by the third feature is relatively high, additional features only have a minor effect. Indeed, adding a third feature turns 15 catchments from failures





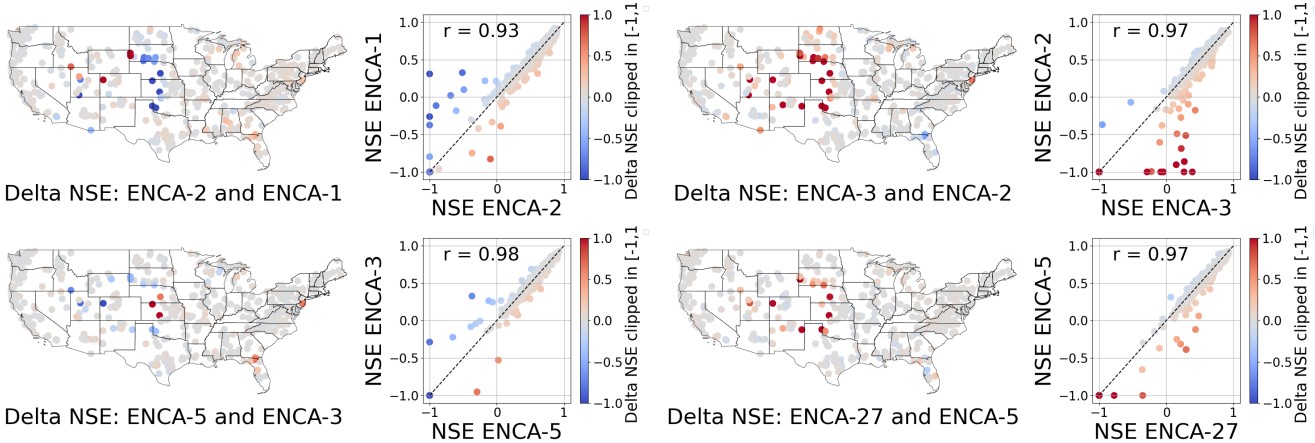

**Figure 5.** Validation NSE of ENCA models with different number of latent features, color-coded according to the NSE difference per catchment clipped in [-1,1]. The improvement of increasing the number of latent features is significant from ENCA-2 to ENCA-3 and marginal for more complex models. The improved catchments are mainly located in the central CONUS, dominated by arid climate conditions.

(negative NSE) to successes (positive NSE) and only leads to marginal deterioration on a few other catchments. Additional features have much less dramatic effects. In order to understand the characteristic of the improved catchments, in Appendix B
we report the performance metrics and some attributes of the 15 catchments failing under ENCA-2, but succeeding under ENCA-3. They are generally characterized by high aridity indexes and intermittent flows, i.e. time-windows in the streamflow time series with low to zero flow, and are for this reason very difficult to predict.

Note, in Figure 2 we computed the GRIDE paths of the ENCA-27 and ENCA-5 latent spaces and concluded that most catchments can be characterized by only three relevant landscape features, while more are only needed for a few special cases.
This discrepancy may arise due to the presence of two or more manifolds with different IDs in the latent space of ENCA. An interesting direction of investigation would be to study this latent space with the ID estimator HIDALGO (Allegra et al., 2020), which allows consideration of multiple manifolds with different IDs.

### 3.2 Interpretation of the Relevant Feature Principal Components

Figure 6 shows the Spearman correlation matrix between the principal components of identified three relevant features and
250 the known streamflow signatures and catchment attributes across different random restarts of the model. The relevant features share information with catchment attributes and hydrological signatures. For instance, feature one carries information about basic hydrological attributes like baseflow index and low flow frequency. Moreover, feature one is (weakly) correlated with soil-related attributes like soil porosity and conductivity, sand, silt and clay fraction. Feature two is correlated with climatic indicators, such as the aridity index, the mean precipitation, high and low precipitation frequency, but also with hydrological
signatures like mean discharge and the $95\%$ quantile of the flow duration curve. We point out that even though the encoder is explicitly designed to learn non-climate landscape features, we can still observe a correlation between latent features and cli-







**Figure 6.** Spearman correlation matrix of the relevant features principal components of ENCA-3 with respect catchment attributes and hydrological signatures across four different random model restarts.



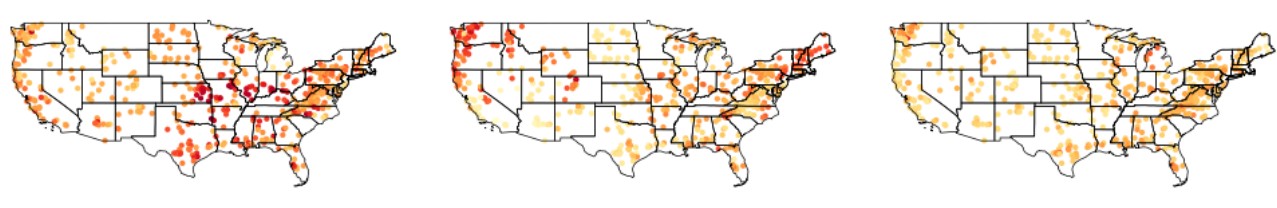

**Figure 7.** Principal components of the relevant features of ENCA-3 in the CONUS.

mate attributes. This correlation can be due to collinearities between landscape and climate attributes. In this case, the collinear attributes are those related to vegetation, like the fraction of forests and the maximum GVF (Green Vegetation Fraction), which are obviously correlated with climate. Finally, feature three is mostly correlated with high and low flow duration and frequency,

signatures relating to the extremes of streamflow. Interestingly, this principal component does not hold much information about neither landscape nor climate attributes, indicating that it encodes catchment information that has not yet been considered or that is not related to any discernible catchment feature. Since the third feature mainly conveys information about certain dry and hard to predict catchments, the latter might very well be the case. The discussed principal components, however, do not share the same amount of explained variance: feature one accounts for about $60\%$, feature two for about $30\%$ and feature three

for about $10\%$. In Appendix E we report the correlation matrices for the other ENCA models, where we can verify that the principal components carry the same information for streamflow prediction consistently across different models. A specific analysis of the correlation between NSE improvements and certain static attributes is provided in Appendix D.

The geospatial distribution of feature importance is shown in Figure 7, where a non-trivial distribution of the features appears, highlighting that the different features have different information content for different regions: feature one dominates in the

270 less to non-arid eastern CONUS, while feature two is mainly dominant in the western part. Feature three does not show such a clear spatial representation. Overall, the potential of delineating geospatial relations is another indicator that the encoder has learnt from the landscape signal in the data.

The elicitation of principal feature components further allows us to pinpoint a subset of the particularly bad performing catchments. These catchments show intermittent flows and they are characterized by relatively high aridity indices (see Ap-

275 pendix B for the catchment characteristic and the hydrographs). Figure 8 depicts the learnt features of ENCA-3, color-coded according to baseflow index (a), aridity (b) and the high flow frequency (c). These are the attributes that show the highest correlation with the learnt features of ENCA-3. The red diamonds represent those 15 catchments failing under ENCA-2, but succeeding under ENCA-3 (see also Table B1). They mainly lie in a sub-region of the latent space characterized by high aridity and low baseflow.





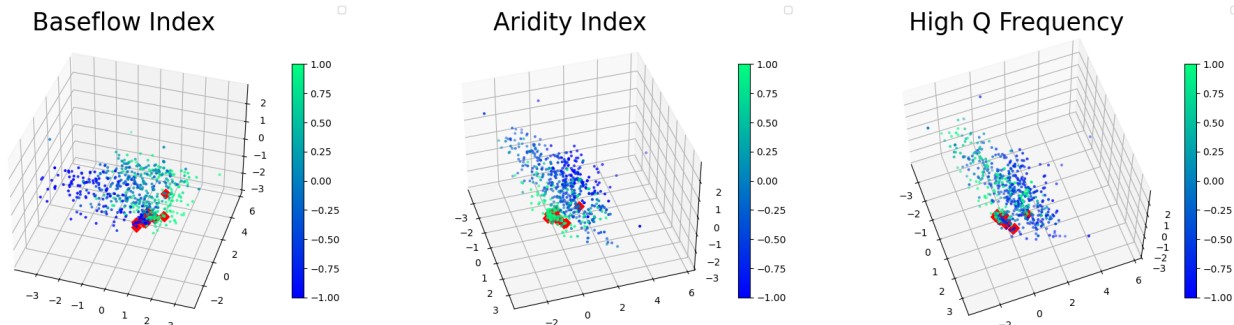

**Figure 8.** Relevant features of ENCA-3 for a random restart, colored-coded according some standardized catchment attributes which correlate strongly with the first three principal components. Red diamonds are those catchments whose prediction has improved from failure (ENCA-2) to success (ENCA-3). These catchments are characterized by intermittent flows.

## 3.3 Relationship between Relevant Features and Parameters in Conceptual Models

In the literature, a great number of conceptual models are available. It is well known that only a handful of parameters can be reliably estimated from the rainfall-streamflow data (Jakeman and Hornberger, 1993), indicating that models with a few parameters and states, like the GR3J (Edijanto et al., 1999) and its successor GR4J (Perrin et al., 2003), must capture the main features of a hydrograph through their structure and parameters. As a consequence, in makes sense to try to relate the parameters of such models to the learnt features identified in this work.

The GR4J model has only four parameters that can be related to specific characteristics of a hydrological system: (i) the maximum capacity of the production store; (ii) the groundwater exchange coefficient that influences the catchment mass balance; (iii) the maximum capacity of the routing store; (iv) the time base of the unit hydrograph that controls the time lag between rainfall and streamflow.

It is conceivable that the number of parameters that need to be calibrated to a specific catchment should be similar to the number of non-meteorological features in the runoff and thus to the minimal number of features needed by ENCA to make good runoff reconstructions. Although a one-to-one mapping may not be possible, we can try to connect these parameters to the relevant learnt features: The first GR4J parameter (maximum capacity of production store) could be related to the second feature (vegetation attributes) since it relates to how much water actually ends up in storage or streamflow and how much is evapotranspired. This threshold parameter in the rainfall-runoff relationship has been related to the root zone and to vegetation indicators (obviously also related to climate) by previous work (Gao et al., 2014). The first relevant feature can be related with the third GR4J parameter that relates to the routing store capacity. In particular, the routing store capacity in GR4J mainly affects streamflow recessions. The first relevant feature is related to baseflow, therefore arguably related to a similar fingerprint the hydrograph. Soil-related attributes have been traditionally related to baseflow (Gnann et al., 2021) which underpins the relation to subsurface storage.



Including the third ENCA-found feature improved performance notably. Meanwhile, it could only weakly be related to static catchment attributes showing some correlation to high and low flow hydrological indices regarding frequency and duration, and to vegetation attributes in Figure 6. Further, including this feature generally helped minimizing the gap between baseflow predictions and observations as it is shown in Figure B1. While the former correlation relates to timing in the hydrograph, the

305 latter two points refer to the water-balance or hydrograph magnitude.

Hence, we conjecture that feature three relates to the two other parameters in GR4J: time lag and groundwater exchange coefficient. First, the third feature results in a smoothing of hydrographs and buffering of discharge spikes (as indicated in Figure B1). In GR4J, the time lag parameter has a similar effect, in other hydrological models this is sometimes referred to as a routing parameter. Hence, the third feature appears to play a role in regulating the timing of increase and decrease of the memory

states of the LSTM. It is noteworthy that these hidden and cell states of the LSTM essentially resemble the role traditionally held by reservoirs in conceptual models and that any routing routine or time lag parameter in conceptual hydrological models acts as a convolutional filter to match the reservoir output to the observed hydrograph. Second, we conjecture that it accounts for offsets in the water balance (see Figure B1) potentially due to water exchange with other catchments. Such a relaxation of a strictly enforced water balance for a modelled catchment further improves the model performance. Interestingly, it was

shown that when using LSTMs for streamflow prediction a strictly enforced water balance deteriorates model performance (Frame et al., 2022). From a hydrological perspective this makes sense: even if the surface delineation of a catchment might be well known, the subsurface delineation might not be identical and unknown exchange fluxes may occur. Further, the observed water inputs and outputs of the system are per se subject to uncertainty and therefore a full closure of the water balance is not guaranteed. LSTMs appear to account for resulting offsets which also holds for our LSTM-ENCA.

Overall, these potential resemblances illustrate how the learning mechanism encapsulates distinctive hydrological characteristics embedded in the model's parameters, making them at least partially interpretable.

## 4 Conclusions

We employed a new kind of autoencoder to distill a minimal set of streamflow features (signatures) necessary for streamflow reconstruction in conjunction with meteorological data. Thus, these features can be interpreted as landscape fingerprints on the streamflow. We compared these features with known catchment attributes in terms of their capacity for streamflow

reconstruction. The primary conclusions we highlight in this study are:

- ENCAs (Explicit Noise Conditional Autoencoders) perform better in terms of NSE and BIAS than the reference attributes enhanced model (CAAM) when the number of latent features is greater than two. In fact, two features seem to be sufficient for most catchments, while a relatively small number of catchments, mostly located in the central CONUS,

require a third one. Including more than three features, however, only leads to marginal improvements. We therefore conjecture that most of the information contained in the static attributes used for CAAM, insofar as it is relevant for streamflow prediction, can be reduced to two independent features. The third latent feature, however, seems to encode information that is not fully contained in those static attributes.





- The correlation between attributes and importance of the relevant features (see Figure 6) suggests an ordering of the information contained in the features for accurately predicting discharge: first, basic hydrological attributes like baseflow and soil-related attributes, followed by the average streamflow and the $95\%$ flow quantile (correlated to climate due to collinearities with vegetation-related attributes) and, third, specifics on the high and low flow, i.e. the extremes of the hydrograph. Looking back at Figure 4, this last feature appears to encode the information that is needed to exceed the model performance that is only based on the 27 static attributes (CAAM).

In summary, our research reveals a significant reduction in the dimensionality of the streamflow time series. Despite the plethora of hydrological signatures and catchment attributes at our disposal, only a small subset proves essential for accurate streamflow prediction. This finding echoes established results from prior studies (Jakeman and Hornberger, 1993; Edijanto et al., 1999; Perrin et al., 2003), suggesting that hydrological systems might be effectively modelled using only a limited set of parameters. The low dimensionality of the relevant catchment information opens up the opportunity for a better *understanding* of its nature, suggesting some future research directions:

- A promising approach could be the adoption of NeuralODEs (Höge et al., 2022), which offer a high level of interpretability due to their low number of states. This combination of a few states and a few features may help to decipher not only the nature of the relevant catchment information but also how it influences streamflow.

- Preliminary analysis (not shown in this paper) has revealed that the known static catchment attributes live on a low-dimensional manifold, which is in line with our finding that only two independent features seem to capture most of the information that is relevant for streamflow. While the correlation-based analysis presented in this paper gives some clues as to how these features can be interpreted, more sophisticated types of analysis like those based on Information Imbalance (Glielmo et al., 2022) might allow for a more precise understanding of their physical nature.

*Code and data availability.* The US-CAMELS dataset, as well as the catchment attributes, is available at the site https://ral.ucar.edu/solutions/products/camels. All the code used for this work is publicly available at the site https://github.com/abassi98/AE4Hydro.





## Appendix A: Validation Metrics

We report some summary statistics of the NSE, BIAS and LOG-STDEV values across the 568 catchments considered in this study. Table A1 shows that the greatest performance gap in terms of the NSE is obtained for the mean, the minimum and the 5% quantile between ENCA-2 and ENCA-3, indicating that the third latent feature is needed to improve prediction in particularly difficult catchments. A similar pattern is shown by the BIAS, whereas the biggest gap is found in the mean, maximum and 95% quantile. Instead, we observe a general tendency of ENCA models to underestimate the flow variability.

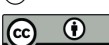



|  |  | CAAM | ENCA-0 | ENCA-1 | ENCA-2 | ENCA-3 | ENCA-5 | ENCA-27 |
|---|---|---|---|---|---|---|---|---|
| **NSE** | Mean | 0.48 | 0.06 | 0.43 | 0.43 | 0.54 | 0.55 | 0.59 |
|  | Min | -23.78 | -71.46 | -18.62 | -22.45 | -10.66 | -9.61 | -6.50 |
|  | Q5 | -0.05 | -0.61 | -0.03 | -0.10 | 0.11 | 0.13 | 0.16 |
|  | Q25 | 0.54 | 0.42 | 0.48 | 0.51 | 0.52 | 0.54 | 0.55 |
|  | Median | 0.69 | 0.60 | 0.65 | 0.67 | 0.68 | 0.68 | 0.70 |
|  | Q75 | 0.77 | 0.73 | 0.74 | 0.76 | 0.76 | 0.77 | 0.78 |
|  | Q95 | 0.85 | 0.83 | 0.85 | 0.85 | 0.85 | 0.86 | 0.86 |
|  | Max | 0.91 | 0.90 | 0.92 | 0.89 | 0.92 | 0.92 | 0.93 |
| **BIAS** | Mean | 0.22 | 0.37 | 0.29 | 0.32 | 0.17 | 0.14 | 0.09 |
|  | Min | -0.55 | -0.66 | -0.82 | -0.67 | -0.69 | -0.59 | -0.61 |
|  | Q5 | -0.22 | -0.23 | -0.27 | -0.24 | -0.26 | -0.26 | -0.25 |
|  | Q25 | -0.05 | -0.02 | -0.05 | -0.06 | -0.08 | -0.10 | -0.10 |
|  | Median | 0.07 | 0.12 | 0.09 | 0.05 | 0.04 | 0.01 | 0.00 |
|  | Q75 | 0.24 | 0.38 | 0.31 | 0.26 | 0.21 | 0.18 | 0.14 |
|  | Q95 | 1.16 | 1.81 | 1.52 | 1.66 | 0.87 | 0.94 | 0.60 |
|  | Max | 6.03 | 15.26 | 11.12 | 15.63 | 8.64 | 7.16 | 6.85 |
| **LOG-STDEV** | Mean | 0.82 | 0.78 | 0.76 | 0.75 | 0.79 | 0.80 | 0.86 |
|  | Min | 0.09 | 0.08 | 0.07 | 0.06 | 0.09 | 0.09 | 0.10 |
|  | Q5 | 0.23 | 0.19 | 0.23 | 0.18 | 0.27 | 0.20 | 0.32 |
|  | Q25 | 0.60 | 0.51 | 0.56 | 0.56 | 0.60 | 0.59 | 0.67 |
|  | Median | 0.78 | 0.70 | 0.72 | 0.73 | 0.75 | 0.75 | 0.81 |
|  | Q75 | 0.95 | 0.87 | 0.88 | 0.88 | 0.91 | 0.88 | 0.96 |
|  | Q95 | 1.44 | 1.47 | 1.36 | 1.33 | 1.33 | 1.38 | 1.52 |
|  | Max | 7.23 | 10.95 | 3.81 | 4.03 | 4.26 | 4.66 | 3.72 |

**Table A1.** Metrics comparison for different models. We report the mean, the minimum, the 5% quantile, the 25% quantile, the median, the 75% quantile, the 95% quantile and the maximum values of the distribution of validation values for the three metrics considered in this work.





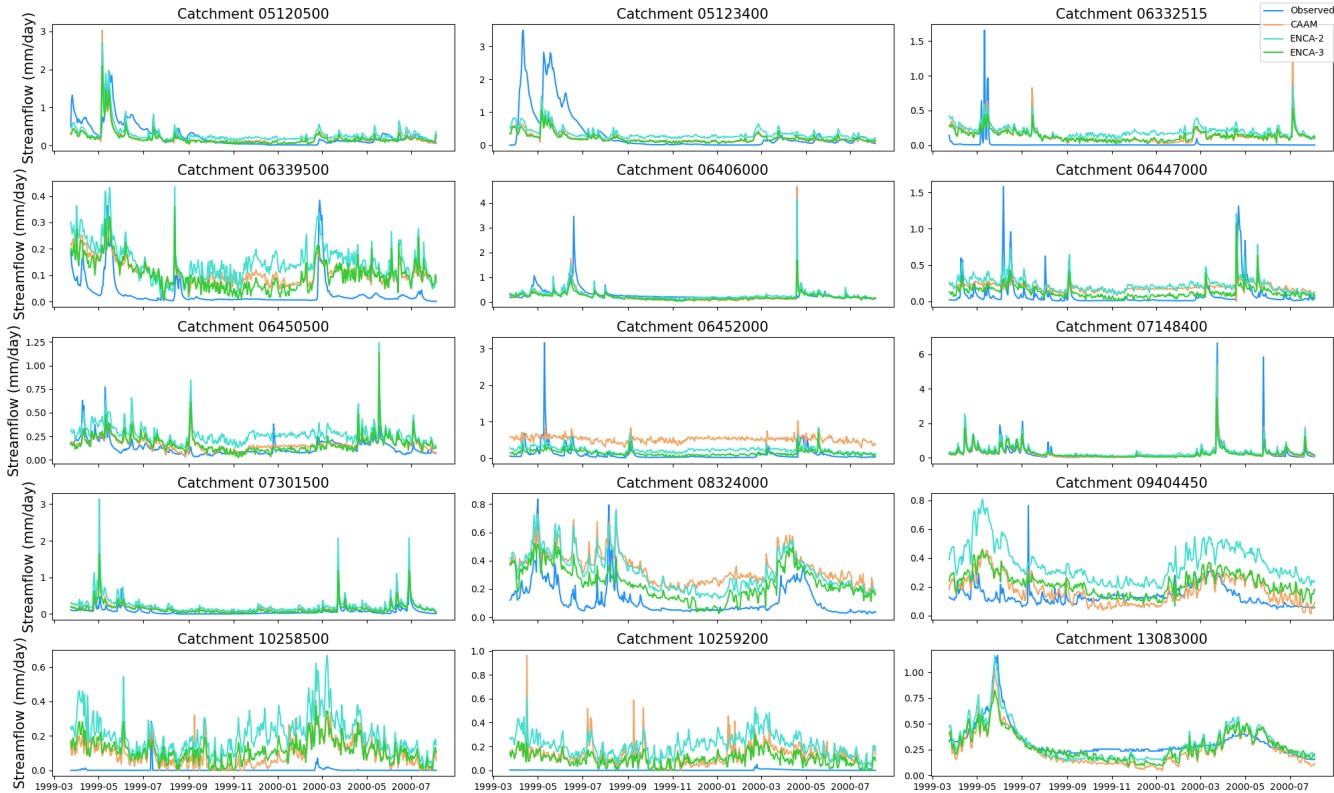

**Figure B1.** Observed and predicted (CAAM, ENCA-2 and ENCA-3) hydrographs for those 15 catchments whose NSE is improved from negative (ENCA-2) to positive (ENCA-3).

## Appendix B: Analysis of Improved Catchments from ENCA-2 to ENCA-3

In Figure B1 we report the hydrographs of the catchments failing under ENCA-2 (i.e. with NSE values below zero), but
succeeding under ENCA-3 (i.e. with NSE values above zero). We report the corresponding validation metrics (NSE, BIAS and LOG-STDEV) and some catchment attributes in Table B1.





| Catchment ID | Baseflow Index | Aridity Index | Q95 | High Flow Frequency | BIAS | | | LOG-STDEV | | | NSE | | |
|---|---|---|---|---|---|---|---|---|---|---|---|---|---|
| | | | | | CAAM | ENCA-2 | ENCA-3 | CAAM | ENCA-2 | ENCA-3 | CAAM | ENCA-2 | ENCA-3 |
| 05120500 | 0.47 | 1.55 | 0.20 | 57.20 | 0.97 | 1.66 | 0.67 | 0.60 | 0.17 | 0.25 | 0.09 | -0.16 | 0.19 |
| 05123400 | 0.44 | 1.44 | 0.32 | 157.35 | 1.41 | 2.01 | 0.72 | 0.11 | 0.06 | 0.09 | -0.00 | -0.27 | 0.22 |
| 06332515 | 0.13 | 2.04 | 0.08 | 34.95 | 3.56 | 4.05 | 2.09 | 0.23 | 0.21 | 0.32 | -0.14 | -0.19 | 0.06 |
| 06339500 | 0.25 | 1.90 | 0.13 | 35.75 | 2.21 | 2.66 | 1.09 | 0.51 | 0.38 | 0.65 | -0.07 | -0.10 | 0.10 |
| 06406000 | 0.70 | 1.66 | 0.33 | 12.35 | 0.52 | 1.00 | 0.49 | 0.64 | 0.37 | 0.67 | 0.24 | -0.04 | 0.25 |
| 06447000 | 0.33 | 2.08 | 0.20 | 33.30 | 2.67 | 2.85 | 1.17 | 0.22 | 0.16 | 0.43 | -0.87 | -0.90 | 0.15 |
| 06450500 | 0.72 | 1.61 | 0.23 | 2.10 | 0.52 | 1.29 | 0.46 | 1.85 | 0.65 | 1.34 | 0.15 | -0.69 | 0.22 |
| 06452000 | 0.43 | 1.74 | 0.28 | 25.00 | 3.41 | 2.03 | 0.83 | 0.27 | 0.29 | 0.70 | -1.94 | -0.49 | 0.16 |
| 07148400 | 0.54 | 1.68 | 0.40 | 13.75 | 0.64 | 1.27 | 0.53 | 1.20 | 0.41 | 0.60 | 0.13 | -0.18 | 0.43 |
| 07301500 | 0.54 | 1.79 | 0.13 | 7.15 | 1.28 | 2.69 | 1.07 | 0.24 | 0.16 | 0.45 | 0.11 | -2.12 | 0.26 |
| 08324000 | 0.64 | 2.00 | 0.59 | 19.90 | 1.35 | 1.71 | 0.93 | 0.71 | 0.57 | 0.71 | -0.25 | -0.86 | 0.27 |
| 09404450 | 0.78 | 2.86 | 0.41 | 3.75 | 0.36 | 1.33 | 0.47 | 0.96 | 0.72 | 0.75 | 0.42 | -1.02 | 0.39 |
| 10258500 | 0.30 | 3.82 | 0.14 | 121.90 | 5.24 | 7.71 | 4.03 | 0.12 | 0.13 | 0.40 | -0.33 | -0.51 | 0.29 |
| 10259200 | 0.27 | 4.76 | 0.17 | 127.00 | 2.54 | 5.37 | 1.92 | 0.20 | 0.11 | 0.44 | 0.34 | -0.01 | 0.23 |
| 13083000 | 0.84 | 2.11 | 0.53 | 0.00 | 0.19 | 0.28 | 0.13 | 1.64 | 1.28 | 1.28 | -0.28 | -0.10 | 0.33 |

**Table B1.** Catchments failing under ENCA-2, but succeeding under ENCA-3. These catchments are characterized by high aridity indexes and intermittent flows.





| Input | Layer name | Hyper-parameters | Output |
|---|---|---|---|
| | streamflow | input | (bs, 5478, 1) |
| streamflow | Conv 1 | 7, 8, BN, Leakyrelu, DR(O.4) | (bs, 5472, 8) |
| Conv 1 | Avgpool 1 | 4 (bs, 1368, 8) | |
| Avgpool 1 | Conv 2 | 5, 16, BN, Leakyrelu, DR(0.4) | (bs, 1364, 16) |
| Conv 2 | Avgpool 2 | 4 | (bs, 341, 16) |
| Avgpool 2 | Conv 3 | 2, 32, BN, Leakyrelu, DR(0.4) | (bs,340, 32) |
| Conv 3 | Avgpool 3 | 4 | (bs, 85, 32) |
| Avgpool 3 | Flatten | N/A | (bs, 2720) |
| Flatten | Linear | BN, Leakyrelu, DR(0.4) | (bs, 512) |
| Linear | Output | BN | (bs, N) |

**Table C1.** The Convolutional encoder architecture used in this study. Batch Normalization (BN) and Dropout (DR) with probability $0.4$ are added between layers. A last BN layer is applied to the decoder output in order to standardize the latent features. $N$ is the number of latent features.

| Hidden size | Initial forget bias | LSTM layers | Dropout | Bi-directional |
|---|---|---|---|---|
| 256 | 5.0 | 1 | 0.4 | False |

**Table C2.** LSTM hyper-parameters. We choose standard hyper-parameters used in the literature (Kratzert et al., 2019).

## Appendix C:  Neural Networks Details

We report the architecture details of the encoder (Table C1) and the LSTM decoder (Table C2) of the ENCA used in this work.



**Figure D1.** NSE difference between ENCA-3 and ENCA-2, clipped in [-1,1], vs some static attributes and signatures.

**Appendix D: Performance Correlation with Known Attributes**

We also report the NSE difference between ENCA-3 and ENCA-2 versus the values of some chosen known static attributes
(Figure D1). We can clearly appreciate a correlation between the performance improvement of ENCA-3 with respect CAAM
for those catchments with baseflow index greater than 0.25, and for aridity indexes greater than 1.0, indicating that the learnt
features are particularly important for improving the prediction accuracy in more arid catchments and for those basins with
greater amount of underground water. A similar pattern is present for the Q95, which can be explained by the fact that autoen-
coders tend to improve the prediction of high flow peaks.



**Figure E1.** Validation NSE of the four random restarts for CAAM (upper left), ENCA-2 (upper right), ENCA-3 (lower left) and ENCA-27 (lower right) for the 568 catchments considered across four random restarts of the model. We do not observe much performance variability across different random restarts of the models.

## Appendix E: Effect of Random Restart

We clearly ascertain that the random restart does not affect much the prediction accuracy (Figure E1). Apart from some catchments, most of them show a consistent behaviour across different random seeds.

We report the correlation matrix between the principal components of the learnt features of ENCA-4 (Figure E2), ENCA-5 (Figure E3) and ENCA-27 (Figure E4) for different random restarts. We notice a consistency across random restarts and different models. Moreover, the correlation becomes weaker and weaker with the fourth component, indicating that 3 features carry most of the information related to streamflow prediction.



**Figure E2.** Spearman correlation matrix of the relevant features principal components of ENCA-4 with respect to the selected catchment attributes and hydrological signatures across four different random restarts of the model.





**Figure E3.** Spearman correlation matrix of the relevant features principal components of ENCA-5 with respect to the selected catchment attributes and hydrological signatures across four different random restarts of the model.



**Figure E4.** Spearman correlation matrix of the relevant features principal components of ENCA-27 with respect to the selected catchment attributes and hydrological signatures across four different random restarts of the model.



*Author contributions.* CA had the original idea and AB and CA developed the conceptualization and methodology of the study. AB developed the software and conducted all model simulations and their formal analysis. Results were discussed and interpreted between MH, CA,
AM, FF and AB. The visualizations and the original draft of the manuscript were prepared by AB, and reviewing and editing were provided by MH, CA, AM and FF. Funding was acquired by AM and CA. All authors have read and agreed to the current version of the paper.

*Competing interests.* At least one of the (co-)authors is a member of the editorial board of Hydrology and Earth System Sciences. The authors also have no other competing interests to declare.

*Acknowledgements.* This research has partly been founded by the SNSF (Swiss National Science Foundation) grant 200021_208249. We
would like to thank Antonio di Noia (Università della Svizzera italiana, ETH Zurich), Fernando Perez Cruz (ETH Zurich), Andreas Scheidegger (Eawag) and Marco Baity Jesi (Eawag) for the insightful discussions.



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
