# Peer review of "Learning Landscape Features from Streamflow with Autoencoders"

_Hydrology and Earth System Sciences, 2024_

## Author Response (AR1)

**Main changes**

- We train all the models for 10,000 epochs and test them on data not previously seen during training (line 137). Test and train loss curves during training are reported in Fig. C1. Even though the models are very big, overfitting can be ruled out.
- In order to further reduce the effect of climate in the learnt features, we condition the ENCA models by feeding known climate attributes as input to the decoder (see, e.g., Figure 1).
- We add the analysis of the three components in which NSE can be decomposed (Eq. 4 of main text of Gupta et al., 2009) in Section 2.3. We comment on low values of NSE obtained (lines 206-214) and the relationship between NSE and its components in Appendix B.
- We move the technical discussion about the Intrinsic Dimension in Appendix A. In the main text, we outline a higher level presentation by using an explanatory figure (Figure 2).
- We report the absolute Spearman correlations of static catchment attributes between themselves (Figure E2). We suppose that learnt features are still correlated with climate attributes due to collinearities, imputable to vegetation attributes (lines 224-227).
- We removed section 3.3, the discussion about possible relationships between learnt features and parameters in conceptual models.
- We have redone all the figures, by taking into account reviewers' comments.

**LEGEND:**
Reviewer comments, Public Answer, Further explanations

**CC1**

Sect. 2.4   Training and Validation

The form  of NSE Nash - Sutcliffe efficiency, Equation (1), is also applied but much earlier by me (Ding, 1974, Equations 40, 47).

In hindsight, this is at variance with the original formulation published by Nash and Sutcliffe (1970, Equation 2) in which the so-called "initial variance" is a constant, as opposed to a model-dependent variable used by both of us.

Duc and Sawada (2023, Equations 3, 24, and Figure 2) recently pointed out, this earliest known variant of the NSE is in fact a standard statistical measure of the coefficient of determination,  $R^2$.  I've had little or no reason to beg to differ, but the authors may think otherwise.

In final analysis, what counts is the base SSE, sum of squares of the error, or its equivalents, the MSE as the authors note in Lines 171-172, or RMSE.

References

Ding, J.Y., 1974. Variable unit hydrograph. Journal of Hydrology, 22(1-2), pp.53-69.

Duc, L. and Sawada, Y.: A signal-processing-based interpretation of the Nash–Sutcliffe efficiency, Hydrol. Earth Syst. Sci., 27, 1827–1839, https://doi.org/10.5194/hess-27-1827-2023, 2023.

The current version is a typo, the numerator should be normalised by observed streamflow as the original version, basically the same as the R2 score.

**Modified this typo. Line 133.**

**RC1**

Review of Bassi et al., Learning Landscape Features from Streamflow with Autoencoders (HESSD)

Bassi et al. use a so-called explicit noise conditional autoencoder (ENCA) to extract a minimal number of streamflow features from a large sample of catchments from the CAMELS US database. The ENCA extracts streamflow features that, together with meteorological variables, allow to reconstruct the streamflow time series. Thus, the features carry information about the way the catchments transform the meteorological signal into the streamflow signal, which should be related to catchment attributes. To explore the information content of these features, the authors relate them to static catchment attributes, hydrological signatures, and (qualitatively) to model parameters of the GR4J model. This shows that the learnt features are related to various hydrological signatures and catchment attributes, especially those related to climate. The authors conclude that the static catchment attributes used contain almost all relevant information to reconstruct the streamflow time series and that these attributes can be summarized only by two to three relevant features extractable by the ENCA.

The paper is well written and of interest to the HESS readership. Overall, I think it has the potential to become a very interesting contribution, but it requires clarifications and revisions before it can be published. Below I list major and minor comments that hopefully help to strengthen the paper.

Thank you !

Major comments:

The authors often write "sufficient", "succeed", etc. I think this is a bit misleading because (a) this binary view is not appropriate when we really look at a range of "goodness of fit" values, and (b) NSE is only one of many possible metrics to assess a model, with often discussed shortcomings (see e.g. Clark et al., 2021).

For instance, in the abstract the authors write "available static catchment attributes compiled by experts account for almost all the relevant information about the rainfall-runoff relationship". But:

(1) The median NSE is never higher than 0.7. That might be viewed as "good" but it is still lower than the maximum value of 1, so there should be room for improvement. (Besides the fact that it is difficult to interpret what a value of 0.7 really means hydrologically for a diverse set of catchments.)

We acknowledge the need for a more cautious interpretation of the NSE values obtained in our study. While these values are generally considered good, it's important to recognize that they may not represent state-of-the-art predictions. The phrase 'available static catchment attributes compiled by experts account for almost all the relevant information about the rainfall-runoff

relationship' should be understood in relation to the results found by ENCA compared with CAAM (the model fed with known catchment attributes). We will revise this statement in the next iteration to provide clearer context and avoid potential misinterpretation.

We stressed that the results are good but not at state-of-the-art level (i.e. Kratzert et al., 2021). Lines 89-90 and 206-213.

(2) The median bias is close to 0, but this should be relatively easy to enforce with calibration. (Also note that most variability in the water balance can already be explained by aridity alone.)

While it's true that bias can be easily enforced to be zero through calibration, our intention in this study was to minimise the imposition of constraints during training to reduce model assumptions. By allowing flexibility in the model without enforcing biases, we aimed to explore the natural variability of the system and avoid potentially biased results. Additionally, it's noteworthy that the influence of aridity alone already explains much of the variability in the water balance, we will investigate this point in future work.

We see that normalized bias is quite good too (Fig.B1).

(3) All models consistently underestimate variability, suggesting that there is some aspect of streamflow dynamics that they miss. (This might also explain why the median NSE is still far from 1, given that NSE should partly relate to how well streamflow variability is captured).

This it is a general feature of training with NSE, that tends to underestimate the flow variability.

We pointed at the relevant literature that present the possible shortcomings of using the NSE. Lines 314-319.

Does that really mean that all the relevant information about the rainfall-runoff process is captured? I would challenge that, because we may not capture many other aspects of streamflow response (i.e. hydrological signatures) that are relevant, for example those related to high flows, recessions, climate sensitivities, etc. At one point the authors even write that "It can partly be attributed to using NSE as objective function for training which puts more weight on matching high flow." So I think some parts of the manuscript need to be reworded to make it clearer that the results mainly refer to a "good" NSE. Generally, I think the manuscript requires a more thorough discussion of possible shortcomings related to the use of metrics such as NSE. It might also be worthwhile to investigate other signatures for calibration and evaluation to get a more differentiated picture of the rainfall-runoff process, so that the statement that all relevant information is captured can be made with more confidence (or perhaps that it cannot be made).

As a side note, the splitting up of NSE (or KGE) into its components – essentially variability, timing, and volume errors – leads to three quite similar metrics. It might be worth considering to use these three components, because they are similar to the metrics used here but their relationship is better understood (see also Gupta et al., 2009). In addition, the other two metrics, especially the variability metric, might be correlated with NSE. It would be interesting to show the correlation between the different evaluation metrics to see how independent they are.

Thank you for your valuable input. Examining the three components in which NSE can be decomposed could indeed provide valuable insights into what ENCA actually learns. We will

enhance the discussion surrounding the NSE values in the next iteration of the manuscript, ensuring a thorough exploration of the limitations and implications. Additionally, your suggestion to consider other evaluation metrics, such as those outlined by Gupta et al. (2009), presents an intriguing avenue for further analysis. We will investigate these metrics and explore their correlation with NSE to gain a more comprehensive understanding of the rainfall-runoff process.

We have reported  the three splitting terms of the NSE and commented on them (Appendix B).. Not all the relevant information is captured within our approach. It is generally difficult to define a good baseline to evaluate the quality of latent features (which is done by comparing the reconstruction NSE). We have rewritten this difficult statement.

My second major point is about what the ENCA really learns. The authors write that forcing is fed to the decoder and thus it only learns "features that are related to landscape properties". Is that really the case? Let's say we feed temperature, potential evaporation (or radiation) and precipitation to the ENCA model (or really any kind of model), then the model somehow represents the interactions between the forcing variables, to some degree mediated by properties of the catchment system (e.g. soils). But: there is a lot of interaction that explains the hydrological response almost independent of any landscape feature. For example, the (long-term and seasonal) interaction between precipitation and potential evaporation can explain most of the variability in the water balance. And the interaction between temperature, radiation and precipitation can explain most of the variability in snow accumulation and melt (at least at the catchment scale). We know this because studies like Knoben et al. (2018) have derived interactive indices that basically take meteorological variables and translate them into "climate fingerprints" that explain a lot of the observed variability in streamflow (though not all of it). In particular, these climate indices explain similar signatures as the ones that are well explained in this study. This makes me wonder to what extent the learnt features are really landscape fingerprints that relate to what we might call catchment form (topography, soils, geology, etc.). Because in Figure 6 it is mainly climate and streamflow signatures that show high correlations, and only a few landscape attributes show weak to medium correlations (and are themselves are related to climate, e.g. forest fraction). I am not an expert in ML methods like ENCA, so perhaps I did not fully grasp the way ENCA works. Still, I would appreciate if the authors could discuss this in more detail in a revised version of the manuscript.

Our hypothesis here is that the decoder, which is an LSTM model, is flexible enough to extract all the information in the meteorological drivers insofar as it is relevant for runoff prediction. This includes all information contained in the interaction between those drivers. Under this assumption, the encoder is incentivized to only extract those features from the runoff that are not stemming from the meteorological data, i.e. landscape features. Of course, this assumption is probably not completely true, and even if it were, the result of the training of the whole Autoencoder might still lead to a certain degree of redundancy, i.e. certain meteorological or climate features might be encoded even though the decoder has direct access to this information. We will discuss these issues more clearly in the revised version. Since we used a uni-directional LSTM, there is the possibility that the decoder does not utilise all the available climate data. Therefore, we are planning to feed this data separately to the decoder (future work).

We report trained models which have been conditioned with climate static attributes too. The test NSE of these models is better with respect to the test NSE of standard ENCA models with the same number of latent features (not shown in the paper). However, the information encoded is surprisingly very similar to those found by standard ENCAs. A possible explanation may be due to the use of a (big) CNN as encoder. Since CNNs consist of several filters, the output of untrained CNN can be biassed towards certain streamflow signatures (like, for example, the high flow frequency).

The discussion around GR4J requires some improvement. It does not actually compare the results to model parameters (either calibrated or estimated based on a priori information), so it remains a qualitative discussion on how certain conceptual parameters relate to the learnt features. It would be much more convincing if some analysis was performed using GR4J to actually show that (a) the GR4J parameters are indeed related to the features (or signatures) discussed (essentially a sensitivity analysis) and (b) whether parameters calibrated to the catchments studied here relate to the extracted features (which would require calibrating GR4J to all the catchments).

We acknowledge the need for improvement in the discussion regarding the relationships between learnt features and conceptual model parameters. To address this, we plan to calibrate the GR4J model using the CAMELS dataset and conduct a correlation analysis between the model parameters and the learnt features. This will allow for a more robust assessment of the relationships between the model parameters and the learnt features, providing valuable insights into the underlying hydrological processes. Thank you for highlighting this aspect, and we will try to incorporate these analyses into our future work to enhance the discussion.

We have already calibrated GR4J augmented with a snow reservoir, which possesses in total 6 parameters. A preliminary analysis shows weak to medium correlations with ENCA's latent features. Since so far the results do not support the claims made in Sec. 3.3, we have removed that section from the paper.

Minor comments

> I was wondering if there is a way to explain more tangibly what the learnt features are, especially for people not familiar with ENCA etc.? How do they relate to signatures (typically single numbers)?
> This is a very good question. So far only correlation analysis, Future work: nonlinear mappings to (i) parameters of conceptual models (ii) known static features.
> Figure 1: estimated Q looks almost like observed Q: is the model so good or is this the same graph?
> Yes it is exactly the same figure, but it is meant to be the reconstructed runoff. We will correct it for the next iteration.
> We have improved the figure 1, in order not to confuse the reader.
> The quality of the figures (e.g. Figure 2) is sometimes a bit poor.
> It should read topographical and not topological in Table 1.
> Yes sorry, we will correct it.
> All figures have been redone.

l.209 and elsewhere: "collected by experts" might be a bit misleading. Many of those attributes are from continental or global maps, which are of course derived by experts (e.g. in soil surveys) but perhaps not with the idea of explaining catchment response.

Maybe better: "known attributes" here .

Modified to "known catchment attributes", everywhere in the text.

Figure 6: are these absolute correlations or why are they >0?

Yes, absolutely. We will clarify it.

Corrected everywhere in the text,

l.222: "does now exceed" should that be "not exceed"?

Yes sorry, we will correct it.

Corrected. We removed that section anyway.

l.230 and elsewhere: I wouldn't say that NSE>0 means "success", see also major comments.

We should write "success a la Kratzert" (Kratzert et al., 2019) where also this definition was used. Otherwise we agree that a NSE of zero does indicate a good performance.

We do not mention at all success or failure now, but only greater or lower than 0.0.

Figure 7: the maps are very small and instead could be shown below each other. They also need to be labelled more clearly, e.g. a, b, c for the subpanels. This is true for many other plots as well.

Figure 8: the axes should be labelled. The standardization also makes it a bit difficult to interpret what we see. Might here be an easier way to show this? May a plot of BFI vs. aridity (potentially coloured according to a third variable), with circles around the 15 catchments?

We apologise for the quality of this and other figures. We will revise them carefully for the next iteration.

Removed this figure, it does not enhance the discussion.

The GR4J groundwater exchange coefficient is a tricky parameter, because it may account for all sorts of water balance problems that might not necessarily relate to groundwater exchanges. This could be discussed somewhere around l.315 that contains a similar discussion on mass balance enforcement in LSTMs.

Yes absolutely !

We got rid of section 3.3

References

Gupta, H. V., Kling, H., Yilmaz, K. K., & Martinez, G. F. (2009). Decomposition of the mean squared error and NSE performance criteria: Implications for improving hydrological modelling. Journal of hydrology, 377(1-2), 80-91.

Knoben, W. J., Woods, R. A., & Freer, J. E. (2018). A quantitative hydrological climate classification evaluated with independent streamflow data. Water Resources Research, 54(7), 5088-5109.

Clark, M. P., Vogel, R. M., Lamontagne, J. R., Mizukami, N., Knoben, W. J., Tang, G., ... & Papalexiou, S. M. (2021). The abuse of popular performance metrics in hydrologic modeling. Water Resources Research, 57(9), e2020WR029001.

**CC2**

This article utilized an explicit noise conditional autoencoder (ENCA) along with meteorological forcings to mimic the time series of streamflow. The model's performance was compared to state-of-the-art models. Subsequently, the relatively small number of features was determined using an intrinsic dimension estimator. Furthermore, the relevant learnt features were correlated with static catchment attributes. Finally, the learned features and GR4J parameters were discussed. It's an interesting work for me. However, I have some comments about this work:

General Comments:

1. Section 3.2: This study showed the Spearman correlation matrix of the principal components of relevant features for ENCA-3 in Fig. 6, as well as for ENCA-4, ENCA-5, and ENCA-27 in Figures E2, E3, and E4. Why wasn't ENCA-2 included? Approximately a dozen catchments were improved from ENCA-2 to ENCA-3, notably those with high aridity indexes (AIs). However, the first and second principal components of ENCA-3 have high Spearman correlation coefficients for almost all of these catchment attributes (baseflow indices, AIs, and Q95). Thus, I am interested in understanding the improvements from ENCA-2 to ENCA-3, which can be interpreted using the Spearman correlation matrix.

While we appreciate your interest in understanding the improvements from ENCA-2 to ENCA-3 through the Spearman correlation matrix, we believe that inserting the correlation matrix for ENCA-2 may not provide additional insights beyond what is already captured by the first two features of ENCA-3. These features exhibit high correlation coefficients with relevant catchment attributes such as baseflow indices, aridity indexes, and Q95. However, to better illustrate the improvements between ENCA-2 and ENCA-3, we suggest referring to

Figures 4 and 5, which present the model performance metrics (e.g., NSE) for both ENCA versions. These figures provide a more direct comparison of the model performance and highlight the enhancements achieved in ENCA-3 compared to ENCA-2. We hope this clarifies the rationale behind our approach, and we appreciate your feedback on this matter.

2. Section 3.3: It's intriguing to draw a comparison between the learnt features and the parameters of hydrologcial models, and the analysis included sounds reasonable as well. However, such discussions appear somewhat subjective. I suggest some additional experiments, such as performing a correlation analysis between the learnt features in this work and the calibrated parameters of the hydrological model, or delving a little further into this component of the work.

Thank you for your insightful suggestion. We agree that further analysis is warranted to better understand the relationship between the learnt features and the calibrated parameters of hydrological models. In future work, we plan to conduct a correlation analysis between the learnt features extracted in this study and the calibrated parameters of the GR4J model.

This comment is related to RC1 and we have already performed experiments that need to be reported.

Specific Comments:

1. Equation 1: The NSE seems like to be different from the original Nash-Sutcliffe Efficiency in Nash and Sutcliffe (1970), of which the $q_{sim,t}$ in the denominator should be $q_{obs,t}$.
Yes it is a typo, it should be the original NSE version.

Corrected, see line 133.

2. Figure 2: The 'Gride' above the subfigures should be 'GRIDE'.
Thank you, noted. We will revise this (in particular) and the other poor figures for the next iteration.

We have redone this figure completely, it is now fig. A1.

3. Figure 3: What do the whiskers of the boxplots mean? 5% and 95% quantiles, or quartile plus 1.5IQR, or alternative interpretations? I suggest making this clear. Besides, The judgement of several models appeared to be primarily concerned with outliers. However, for example, the 25% - 75% quantile of ENCA-2 seems lower than that of CAAM. In this regard, CAAM seems to be more similar to ENCA-3 or even ENCA-5.
Yes the whiskers mean the 5% and 95 % quantile. We will clarify this aspect for the next

iteration.  You are right that the biggest differences in terms of NSE are observed for outliers when the number of latent features is greater than 2. However, from ENCA-1 to ENCA-2 we also observe a neat difference in the bulk, suggesting one learnt features is not enough to encapsulate the information contained in the straemflow.

It is true that ENCA-2 quantiles 25/75 are more similar to ENCA-5. (For the next iteration, it would be maybe easier to visualise a metric distance between NSE distributions instead of the boxplots.)

We added the explanation of the meaning of the whiskers.

4. Line 176: 'asses' should be 'assess'.

Yes thanks, noted.

References

Nash, J. and Sutcliffe, J. (1970). River flow forecasting through conceptual models part i — a discussion of principles. Journal of Hydrology, 10(3):282–290.

**RC2**

*General Comments*

The authors developed a ENCA -based method to extract relevant physio-geographic and climatic catchment features that are still able to well reproduce the observed discharge time series from 568 catchment of the CAMELS data.

The paper is in my opinion highly relevant given the current developments in using KI in Catchment Hydrology and contributes with some novel aspects. It is generally well structured, compactly written losing relevant information and is easy to read. I belief therefore the manuscript is well suited for publication in the HESS journal.

Thank you !

Some comments/suggestion that I believe would improve the manuscript and that should be addressed before final publication is the following:

> In line 119 you define the output of the encoder as "relevant landscape features" – I would mention that it is technically a vector with N numbers (where the derivation of setting/determining setting of N is explain later).
>
> Thank you, we agree that it would make the text clearer to specify that we the encoder learns a vector of features for each catchment.
> Added in lines 115-116.
>
> Concerning the ENCA process – I see the problem/issue that the derived latent features will be strongly influenced by the characteristic (Bias, variability, etc.) of the meteorological data. While in this paper only 1 dataset is used, I would like to recommend a study by Lehmann et al. 2022 who analysed 1600 of different meteorological data products (for Eta, P, Q) in comparison to GRACE data. In order to examine the robustness of the approach presented here – I would like to see how, consistent the latent features will be estimated under different "quality" of data.
>
> Thank you for bringing up this important consideration. Exploring the robustness of the ENCA approach to different meteorological data products, as highlighted by Lehmann et al. (2022), indeed seems like an intriguing direction for future investigation. While we acknowledge the potential influence of meteorological data characteristics on the derived latent features, we currently focus on analyzing the performance and implications of our approach using the dataset available in this study. However, we appreciate your suggestion and plan to explore the robustness of the ENCA method to varying data qualities in future work.
> This is for future work. See also Kratzert et al. 2021, where they use different met inputs.
>
> I would put the technical formulations for the GRIDE estimator in an appendix and replace it by a more illustrative description and figure! The analysis in section 2.3 was difficult to follow.

Yes, we agree with your assessment. The technical formulations in section 2.3 are indeed dense and may be difficult to follow. As suggested, we plan to simplify the explanation and supplement it with a figure to enhance clarity.

We moved the technical GRIDE discussion in the appendices and added an explanatory figure in the  main text, fig. 2.

Gupta et al. 2009 (J. Hydrol. 377, 80-91) have shown, that for optimal NSE estimates, the variance of the observations will be systematically underestimated in the predictions!

Thank you for the reference, we will add it to the manuscript.

See section 2.3 and appendix B.

Section 2.5 – I would like to see a nice graph/figure where the latent feature reduction/followed by PCA and correlation analyses with catchment characteristics is given. Is there any attempt done to analyse any kind of non-linear dependencies between latent features and catchment characteristics?

It is a very interesting question. We plan to address to find a non-linear map between learnt features and known attributes in future work.

We believe this is beyond the scope of this paper and we will leave it for future work.

Why is it so important to reduce on the minimum number of latent features? We see that with each feature the performance is better (even though increasing less, and less). Is it importance of feature authors are interested? What are the consequences now?

We acknowledge that increasing the number of latent features generally leads to improved model performance, albeit with diminishing returns. While there is no strict requirement to minimize the number of latent features, we find it noteworthy that there is a significant improvement in model performance between 2 and 3 learnt features. However, as the number of features (N) increases beyond this point, the performance gains tend to be marginal.

We agree that the cutoff for the number of features is somewhat arbitrary. However, it is interesting to note a strong dimensionality reduction of the learnt features concerning runoff (Q) and the static attributes. This reduction suggests that the model is effectively capturing the most relevant information from the input attributes, leading to improved performance with a relatively small number of features.

It is natural to question why the model selects only 2 or 3 relevant features out of the 27 attributes fed to CAAM. Further investigation into the optimal number of features and their relationship to model performance could provide valuable insights into the underlying hydrological processes and model representation.

Commented on the arbitrarily of the cutoff, line 184.

Minor Changes:

L3: Number and types of signatures - could you 2-3 examples for types of signatures, here.

Sure, we will add them

Table 1 provides a lot of examples for hydrological signatures.

Table 1 is hard to read

We absolutely agree, we will reformat this table.

Made this table more readable.

L214/215: I do not understand this statement

Standardisation is done in two different ways (L129) such that maybe the magnitude of the learnt features fed to the decoder of ENCA is different from that of the known attributes fed to CAAM. This can maybe cause a different reconstructed variability. This is however just an hypothesis and requires more work to be verified.

Figure 7: More heading – what do the colours mean?

We apologise for the lack of labels and colour legend. We will add it for the next iteration.

We have redone this figure.

Figure 8: what does "relevant" mean here?

The relevant landscape features found by ENCA (L119-120)

Table C1: define bs

Yes thank you! We will correct it.

Done.

I feel, the manuscript has in general the potential to be a valuable contribution to HESS, however, questions and issues raised in the general comments would need to be addressed and discussed to a significant part before final acceptance.

Thank you for your comments and feedback !

**RC3**

This contribution explores the creation of learned streamflow representations with an autoencoder based approach. However, I do find the approach very, very cool and do therefore hope that HESS accepts the paper. I do just have a single major point. Alas, I do fear that this point will result in a lot of work for the authors (sorry). Additional, I have a surprising amount of smaller comments. In this regard, I would like to especially point out that the quality of the figure is at time mediocre to bad, and I would wish that the senior author takes more care in proofing these simple things before handing in the manuscript. Reviewers that do not like the basic idea could easily use these superficialities to bombard the overall work. Given these two points I feel obliged to propose a major revisions. But, I do hope that my comments help improve the manuscript so that it becomes HESS ready asap.

Thank you for your comments !

Major comment

My main concern with the paper is the lack of context for the results. Due to the lack of baselines and reference points I was able to understand what the results actually imply. I do not know if it is impossible as such --- but I did read the comments from the other reviews and got the feeling that they had a similar problem (but where not able to address it). Thus, I predict that many readers will be lost. I will now give three examples for what I mean:

*Example 1*: The way the models are setup is rather unique in hydrology and very different from what most readers are currently used to. If I understood what you are doing correctly (which is not easy in the first place). You have around 568 samples of length 15*365=5478 (because you use 15 years of daily data; see: L. 110). For comparison. The most basic machine learning dataset, called MNIST, has 6000 samples of length 28*28=784, and the model trained in Kratzert et al. (2019) is trained in a many-to-one setting resulting in 531*15*365=2907225 samples. Also, the models in Kratzert et al. (2019) are trained for 30 epochs, while the autoencoders in the current publications are trained for 2000 epochs (L. 173).

You are absolutely correct; the architecture we have employed is indeed unique, making direct comparisons with models in existing literature challenging. Our aim is not to outperform state-of-the-art models but rather to extract pertinent features for runoff prediction. To facilitate comparison, we contrast ENCA with a model, CAAM, operating within the same setup but

utilising catchment attributes. Our objective is to gauge the learned features against established attributes documented in the literature, necessitating a metric for runoff reconstruction evaluation. It's worth noting that we employ the same dataset as Kratzert et al. (2019), albeit with differing preprocessing approaches. Kratzert et al. (2019, 2021) augmented the dataset, repeating each sample 270 and 365 times respectively, whereas our models utilise each sample only once. Consequently, we incorporate a fully connected layer at the outset to augment model capacity. In fact, we conducted experiments (not detailed in the paper) revealing that without the fully connected layer, the resulting Nash-Sutcliffe efficiency (NSE) is significantly low, averaging around 0.0.

Further, the autoencoders are models with much more capacity than the models that are commonly used for the reference datasets that I just mentioned. To give a rough estimation: The models in Kratzert et al. (2019) ingest approx. 30 inputs and have a hidden size of 250 yielding roughly 260,000 parameters. In contrast, in the current paper the LSTM has the same hidden size, but ingests around 1350 inputs (neglecting the different latents from the CNN for now), which yields a magnitude more parameters. As matter of fact, roughly: 1,650,000. And here I did not even count the parameters of the fully connected part and the convolutional network.

Now, counting the number of parameters and model updates is seldom useful. However, I do have to admit that I was surprised then about the lack of validation split, because I would expect that models this large trained on so few samples will easily overfit. I did therefore make a comparison run where I roughly used the settings from Kratzert et al. (2021) (which is a bit more up-to-date in terms of approach than the ones from the 2019 paper, since it uses different dynamic inputs) but the 15 years training-test split from the current paper. And, without any hyper-parameter tuning, I got a median NSE of 0.8 (And a Bias of 0.0097) after less than 20 epochs . From what I can tell this is better than any of the models presented in this paper (see e.g., Table A1). However, with the current version of the manuscript readers are not made aware of this. Currently, papers like Kratzert et al. (2021) or Klotz et al. (2022), who show current best performances/practices, are not referred to at all and I fear that most will not know such things (and probably not look in the corresponding reference). Even I had too look up the reference results and train a model on CAMELS to get a feeling for the difference in performance --- and I worked with this data for a long period of time now. I think it would be good to provide such results as points of reference for readers. But, I think you should also go further and check if you have an overfitting problem that might corrupt your results; or at least discuss this potential problem in your methodology.

You are correct in noting that the autoencoders possess significantly more capacity compared to commonly used models for reference datasets. The total number of parameters is indeed  slightly larger than 3 Millions, while the total number of training samples is 531x5478= 2 908 818, making the network slightly over-parameterized, and potentially leading to overfitting. In order to address this problem, we conducted explicit validation across a different time period: training was conducted using data from the first 15 years, with validation performed on the subsequent 15 years using early stopping (with patience of 2000 epochs) with a maximum of 20 000 epochs. This temporal split ensures that the model's performance is evaluated on unseen data. Moreover, the models were regularised with dropout between layers (with probability=0.4) and the training was stopped with early stopping. We did not consider the test split necessary since we did not perform hyper-parameter selection.

However, even if the models may not overfit the training dataset, the models chosen are the ones with best validation loss (or nse). This way, we implicitly introduced a bias. In this case a test split would be therefore necessary to obtain unbiased estimates of the NSE values. Nevertheless, we already conducted experiments where the models selected are the last trained models after 10 000 epochs (thus without relying on the validation split and similarly to what was done by Kratzert et al., 2019). This second approach shows similar results to the first one, since the training/validation split was the same and the models converged around 10 000 epochs also in the first approach. This way, the test split is no longer necessary.

Note, however, that this good but low NSE values (compared to state-of-the-art approaches) can also arise due to the long sequences fed to the LSTM decoder, which in our case is 5748, while in the referenced literature this number is significantly lower. We explicitly selected such long sequences in order to extract long-term climate information that cannot be detectable when the fed sequences are on the order of one year (Kratzert et al, 2021) or less (Kratzert et al, 2019).

Training is performed for 10000 epochs and the last model is chosen for testing. There is thus no more need for a validation split. We reported the plot of the MSE values during training on the train and test split in the appendices, see training curves (Fig. C1). The discussion on the NSE values obtained is at lines 26-214.

*Example 2*: In Figure 6 you show a correlation matrix between the PCA components of ENCA-3 and the catchment attributes. These attributes include the hydrological statics that Kratzert et al. (2019) and you did not use (as a side note: I am not sure if the runoff ration and the streamflow elasticity should be considered climate features). The reason that Kratzert et al (2019) did not use these features is that the model therein is supposed to be a simulation model with the capacity to work everywhere --- even in ungauged settings where such information is not available. I do, however, argue that the current version of the manuscript does not tell readers enough gauge their importance. First: if you argue that the learned features encompass the hydrological attributes then it would be necessary to have references that show how models trained with these attributes actually perform (this can be easily shown by showing the performance of one reference with all the attribtues (including the hyrological ones) and another with just the hydrological ones). Second: To me it seems that the correlation of the PCA components with the static attributes is not enough to judge anything, since I do not know how the static attributes are correlated within themselves. For example: I would expect that is correlated with the Low Q Frequency or the runoff-ratio with the Q Mean. Alas, this is not shown in any figure and readers are in the unsatisfying situation that they have to guess, compute these themselves or (and I fear this will happen to most) will remain ignorant of this aspect of the analysis. The remedy is also so very simple: Show the spearman correlation matrix of statics with themselves as a point of reference.

Regarding the placement of runoff ratio and stream ELAS, we acknowledge your point. Indeed, they share characteristics of both climate attributes and hydrological signatures.

In table 1 the reader can find these attributes among the hydrological signatures.

We argue that learnt features encompass catchment attributes (which do not include hydrological signatures). We utilise the hydrological signatures solely for the purpose of comparing the learned features and do not incorporate them into the training of our Catchment Attribute-Augmented Model (CAAM).

We stress that the hydrological signatures have never been included as input in our trained models. Rather, we report them to get a comparison with our learnt features. Learnt features steam from the streamflow and they are technically machine learning equivalents of hydrological signatures (since they are functions of the runoff). However, by conditioning the encoder on climate variables, the goal is to ideally remove all time-dependent forcing information, such that what remains can be assimilated to landscape attributes. In the language of Approximate Bayes Computation (ABC) for stochastic models, these learnt features are called summary statistics, because they contain information about a model's parameters that ideally contain no noise (see Albert et al., 2022). Therefore, it is natural to compare learnt features with models that do not see hydrological signatures in the input. Adding known streamflow signatures to the benchmark does therefore not lead to new insights.

Additionally, your suggestion regarding the correlation matrix of static attributes with themselves is well-taken. We agree that this information is crucial for a comprehensive understanding of the analysis. To remedy this, we will include the Spearman correlation matrix of static attributes in the appendix of the revised manuscript, providing readers with a clear reference point.

We have added the correlation between catchment attributes themselves in the appendix D, to make the reader aware of them and possibly explain the amount of correlations still present between learnt features and climate attributes.

Minor comments

*Figures*

Please adjust the the font-size of all figures. I am not saying that all figures need to have the same font-size (that would be pedantic), but some of the figures are plotted with font-size that seems to be made for posters or presentations (e.g., Fig. 6 or Fig. E1), while others are very difficult to read on A4 (e.g., Figure B1). On top of that some Fig. 2 and Fig. D1 have very bad quality and should be provided in higher resolution. I will now go through all figures of the main manuscript in detail, but I'd like to mention that the ones in the Appendix would also benefit from some love.

Done

Figure 1. This is a great figure. I really like it and it helped me to understand the setup greatly. As a matter of fact, it is the only figure I have nothing to complain about.

Thank you !

Figure 2. Three things: (i) I think it would be good to only use a single figure heading here. The right-hand side of the figure does not profit from having "Gride Evolution" written on top of it. (ii) In the main text you write "GRIDE", but here you use "Gride". (iii) Please show all restarts and ENCA models. The plateau that you mention the main text is only really visible for ENCA-27 (left-plot) and there is enough space to show the full information if you make the plot a bit smaller. Or, if you do not like that you could provide the additional plot the appendix (like you do with the correlation matrices). As it is now readers have to trust your statement (and I am sure it is true), when you could just show it.

You are absolutely right! We will include all the restarts in the updated version.

Done, move to the appendix, fig A1.

Table 1. The whole table is extremely wrapped and difficult to read. I think its worth to format it nicer and maybe use two pages if necessary or move it to the Appendix.

Thank you, noted ! It will be improved for the revised manuscript.

Done, splitted in two pages

Figure 3. I like to propose to add vertical line between the reference models and the ENCA-n (with n > 0) models. That way the plot would become visually much clearer.

It is true, we will add it in the revised version.

Done

Figure 4 and Figure 5. I think the coloring is off. The figure caption says that the color indicates the difference per catchment clipped in [-1,1]. Thus, grey points should be on the 1:1 line. However, at the clipping position s they are not. For example, in Figure 4 the point at [-1,-1] is colored in dark red (indicating that ENCA-3 performs much better than CAAM for this basin). My believe that the coloring happens before the clipping here, however this is nowhere described so I cannot be sure.

Yes you are right, in Figure 4 we first colour and then clip. We will explain that more clearly in the revised version.

We have reported the the clipped nse colored with the clipped difference (these clipping are done independently)

Figure 6. See Major comment.

Figure 7. It is unclear what this is. There exist principal components for multiple restarts and readers do not know what restart from Fig. 6 is used here (or whether this is a new run). The figure itself also does not explain what the color-coding is which maps shows which component. I can, of course, infer that the coding is the magnitude of the component (without showing the

value range), and the ordering is from left-to-right); but this should not be a leap f faith on part of the reader.

You are right ! They are the principal components (ordered from the left) of the first restart. We agree that this figure needs the labels and the colour-coding explanation. We will add it into the revised version of the manuscript.

Done. Now we have the relevant features for all the restarts.

Figure 8. It is difficult to see anything from this figure.

Removed.

*Text*

L. 38-39. Isn't the current analysis also based on predefined model assumptions (just less so)? Like, e.g., that a CNN is appropriated to encode the runoff and that an LSTM is appropriate as decoder.

Removed that phrase.

L.49. Technically the first study shows that data-driven approach outperforms process-based model in prediction accuracy. No "might" needed.

Right, noted !

Done

L. 50. becuase -> because

Done

L. 50-51. I disagree that sure if the results from Kratzert et al. (2019) suggest that information in the catchment attributes is what was previously not utilized fir streamflow predictions.

We mean that catchment attributes encapsulate information not previously utilised in conceptual models.

L.56. I do not understand what is novel about the architecture. The architecture itself seems to be just a conditional auto-encoder. Maybe I am missing something here though (in which case it would be good if you explain it somewhere).

Rephrase: first time used in hydrology

Done

L.58ff. I argue that it is unclear at this point what a "stochastic model simulator" is. I had to rad the referenced publication by Albert et al. (2002) to figure that out. Therein the system innovations are noise, so I kind off see, how it makes sense. In this study, however, the system innovations

are given by meteorological forcing instead of noise. As I see it is not necessary to use this terminology here and the whole paragraph can be shortened, simplified, and made clearer without it. If there is no good argument which I missed, I would therefore propose to avoid it here and then introduce it only in section 2.2.

Ok, noted !

I think it is clear, nothing to do.

L.70. You do not benchmark your models at all.

Against CAAM.

Modified to compare.

L. 70-71. Long Short Term Memory -> Long Short-Term Memory

Thank you, noted !

Done

L.90 I do not understand the reference to ungauged basins in the first place. Why not use references to gauged basins that use a time-split like you do in this manuscript? Especially, since in that setting it has also been shown that LSTMs tend to outperform classical approaches.

You are right, it is a typo that we will rephrase.

Removed that reference.

L.91-92. For me, the claim that the resulting knowledge might become useful for ungauged catchments in the future would need a bit more discussion for me, since your approach assumes knowledge of whole hydrographs to synthesize static attributes.

Thanks for your valuable insight. With this approach we can learn which attributes are more important and then use them in the ungauged setting. But we agree than it needs more explanation.

Removed that reference.

L. 111 The statement that the memory cells of the LSTM are limited by the dimension of the input layer is simply wrong. As a matter of fact, the dimension of the input layer has no influence on the number of memory cells. The hyper-parameter that defines the number of memory cells is the "size of hidden sates" or "hidden size". Table C2 shows that you have an LSTM hidden size of 256. Hence, this study uses LSTMs with cell states vectors of size 256.

You are right. Sorry for the confusion, we meant the network capacity.

Modified to capacity

L. 117. I find this not so obvious, given that you use a data-split with validation (hence the training data can be memorized without learning to actually hand over the required information) and that

the LSTM has to learn to push this information through it precisely. Checking this claim would be an interesting experiment for the Appendix and provide a useful point of reference for readers.

It is indeed interesting and we will try to add it into the next version.

Now refer to a theoretical reconstruction of the entire time series. However, it may not be true since we have convolutional filters.

L. 132. I think it would be interesting if you report the batch statics somewhere (and perhaps compare them to the statics of the attributes).

What do you mean by batch statics?

We have added them for ENCA-5 (Table C2).

Section 2.3 I would argue that GRIDE is insufficiently described here. I had to look up the original publications to see what you are doing. I would propose to either describe it on a much higher level, where you try to explain the basic idea of it (and then not introduce the additional notation and perhaps even move Figure 2 to the appendix); or to describe it in more detail so readers learn about the method from the current papers itself (still, I would then suggest to do it without the proofs).

Moved the technical discussion to appendix and left only a high-level discussion in the main text.

L. 216. I think Kratzert et al. (2019) is not a good reference here, since those results are about ungauged catchments, while here we discuss a gauged setting.

In Kratzert et al (2019) there are however a lot of results regarding the gauged settings, what were called "global" models.

I checked the references with Kratzert et al (2019) and tehy seem consistet to the message we want to present here.

L. 237f. Would be good too see if this pattern is consistent over more than just one model run. But, I can understand if you do not want to make that effort...

Thank you for your comment. Could you please be a little bit more specific on the clarification needed?

All the figures now refer to more model runs to show consistency.

L. 252-253. In Fig.6 feature 1 does not seem to be much better correlated to soil attributes than to Climate attributes. Maybe adding the actual correlation values would help readers to see what you mean here.

Thank you for your comment. It would indeed help, even if however should be already clear from the colour. We will add the actual values for more clairity.

Done, added mean plus minus std.

L. 257ff. Please just show whether this collinearities exist! This can be easily checked, no? It should not be a point of discussion, but a point of analysis.

You are right, we will analyse this collinearities.

Done, lines 225-227.

Sect. 3.3. Interesting discussion.

Removed this section.

Sect. 4. It might be good to call the section "Conclusions and outlook" or something similar to accommodate the second part of the section. Else, I really liked the section.

Agree, we will rephrase
Done.

---

## Author Response (AR2)

**LEGEND:** Reports Answers

**Report #1**

Overall, the authors responded well to most of my comments and I think the paper can be published after some minor revisions (see below). Congratulations on a great paper!

Thank you for your comment and support throughout the whole revision process!

I have a few more comments that are related to comments I made in my previous review.

I am still not entirely convinced that "landscape features/properties/fingerprints" is the best choice of wording, e.g. in the abstract or line 117 or 244 (of the revised version). What the algorithm learns from the streamflow data that is not contained in the raw climate data is not necessarily (or at least not only) related to landscape features in the sense of soils, geology, etc. I would rather call it "hydrological features" or something similar, which are both a result of the interaction between the climate variables themselves and the landscape. This matches well the fact that they are strongly correlated with hydrological signatures like baseflow index or mean streamflow (the latter being mostly a signature of the climate).

Thank you for your comment. It is true that the learnt features are technically hydrological signatures (and are clearly correlated to them) because they are functions of the streamflow. Moreover, they must be minimally related to climate, through the conditional structure of the architecture. However, this holds true only assuming a perfect decoder, i.e. a decoder capable of utilising all the relevant information (relative to streamflow reconstruction) about the meteorological drivers. If such is the case, all the interactions between climate and landscape would be internally represented by the decoder, which would "know" how to merge the information coming from the encoder. The encoder would be then free to learn any additional information contained in the streamflow which is not contained in the drivers, and this information would be related to landscape properties not influenced by the climate. Of course, known landscape attributes are something different because they also contain climate information.

The conclusion that "our research reveals a significant reduction in the dimensionality of the streamflow time series" hinges on the choice of objective function (i.e. NSE). I mentioned in my previous review that NSE does not capture all that is hydrologically relevant (that's the whole point of using diagnostic signatures in the first place). Thus, these two to three features might be enough to achieve a high

NSE, but it does not mean that it captured all information contained in the streamflow time series. I think that should be clarified in the conclusions.

Thank you for your comment. Of course, the learnt features itself and so the information that they can capture depend on the specific calibration function utilised. We will specify it in the conclusions. Any calibration function has certainly advantages and disadvantages, and none is perfect at all tasks. Moreover, the definition of what is hydrologically relevant is somewhat subjective. Nevertheless, in this work we have empirically shown that the performance of ENCA models is satisfactory in all the terms in which the NSE can be factored and are traditionally considered hydrologically relevant. However, we believe that a comparison between different calibration metrics is an interesting avenue of exploration, but it lies outside of the purposes of this work.

Some minor points:

In the abstract, it still says "collected by experts", which I previously suggested to be changed.

Thank you for spotting this typo, we will modify it.

In Table 1 it should read topographic, not topologic. It would also be good to report all units.

Thank you again for spotting this other typo, we will correct it.

**Report #2**

From a reviewer perspective I am extremely happy with the result. From the start on the authors had a good core idea To summarize the paper a bit too much: The underlying information of landscape features for rainfall--runoff modeling lives on a small dimensional manifold. I really liked this idea from the start and I am glad the authors put so much work into the manuscript to polish it. In short: The paper started well, but throughout the peer review process the authors managed to improve every aspect of the manuscript and sharpen its focus!

Thank you for your comment and support throughout the whole revision process!

---

## Author Response (AR3)

**LEGEND:** ==Reports== Answers

**Report #1**

Thanks again for the responses and revisions made by the authors. The paper can essentially be accepted as is, but I have two technical suggestions.

Thanks again for all the revision work !

The line numbers refer to the track changes version.
l. 120: "contain minimal information" - not sure if "minimal" is the right wording here. Do the authors mean minimal in a sense of "most compressed", since hydrological signatures reduce some aspects of a hydrological time series to a single number?

We mean here that the learnt features must contain as little information as possible about the climate and the drivers, due to the architecture we chose (modified in line 117 of the revised manuscript).

l. 123 replace "e" with "a"

Thank you for spotting this typo (we corrected it in line 120 of the revised manuscript).